RESEARCH ARTICLE  

# Collaborative hunting in artificial agents with deep reinforcement learning

Kazushi Tsutsui[1,2]*, Ryoya Tanaka[2,3], Kazuya Takeda[1,4], Keisuke Fujii[1,2,5,6]

[1]Graduate School of Informatics, Nagoya University, Nagoya, Japan; [2]Institute for Advanced Research, Nagoya University, Nagoya, Japan; [3]Graduate School of Science, Nagoya University, Nagoya, Japan; [4]Institute of Innovation for Future Society, Nagoya University, Nagoya, Japan; [5]RIKEN Center for Advanced Intelligence Project, Tokyo, Japan; [6]PRESTO, Japan Science and Technology Agency, Tokyo, Japan

**Abstract** Collaborative hunting, in which predators play different and complementary roles to capture prey, has been traditionally believed to be an advanced hunting strategy requiring large brains that involve high-level cognition. However, recent findings that collaborative hunting has also been documented in smaller-brained vertebrates have placed this previous belief under strain. Here, using computational multi-agent simulations based on deep reinforcement learning, we demonstrate that decisions underlying collaborative hunts do not necessarily rely on sophisticated cognitive processes. We found that apparently elaborate coordination can be achieved through a relatively simple decision process of mapping between states and actions related to distance-dependent internal representations formed by prior experience. Furthermore, we confirmed that this decision rule of predators is robust against unknown prey controlled by humans. Our computational ecological results emphasize that collaborative hunting can emerge in various intra- and inter-specific interactions in nature, and provide insights into the evolution of sociality.

## Editor's evaluation

Cooperative hunting is typically attributed to certain mammals (and select birds) which express highly complex behaviors. This paper makes the valuable finding that in a highly idealized open environment, cooperative hunting can emerge through simple rules. This has implications for a reassessment, and perhaps a widening, of what groups of animals are believed to manifest cooperative hunting.

*For correspondence:
k.tsutsui6@gmail.com

Competing interest: The authors declare that no competing interests exist.

## Introduction

Cooperation among animals often provides fitness benefits to individuals in a competitive natural environment (*Smith, 1982*; *Axelrod and Hamilton, 1981*). Cooperative hunting, in which two or more individuals engage in a hunt to successfully capture prey, has been regarded as one of the most widely distributed forms of cooperation in animals (*Packer and Ruttan, 1988*), and has received considerable attention because of the close links between cooperative behavior, its apparent cognitive demand, and even sociality (*Macdonald, 1983*; *Creel and Creel, 1995*; *Brosnan et al., 2010*; *Lang and Farine, 2017*). Cooperative hunts have been documented in a wide variety of species (*Lang and Farine, 2017*; *Bailey et al., 2013*), yet 'collaboration' (or 'collaborative hunting'), in which predators play different and complementary roles, has been reported in only a handful of vertebrate species (*Stander, 1992*; *Boesch and Boesch, 1989*; *Gazda et al., 2005*). For instance, previous studies have shown that mammals such as lions and chimpanzees are capable of dividing roles among individuals, such as when chasing prey or blocking the prey's escape path, to facilitate capture by the group

**eLife digest** From wolves to ants, many animals are known to be able to hunt as a team. This strategy may yield several advantages: going after bigger preys together, for example, can often result in individuals spending less energy and accessing larger food portions than when hunting alone. However, it remains unclear whether this behavior relies on complex cognitive processes, such as the ability for an animal to represent and anticipate the actions of its teammates. It is often thought that 'collaborative hunting' may require such skills, as this form of group hunting involves animals taking on distinct, tightly coordinated roles – as opposed to simply engaging in the same actions simultaneously.

To better understand whether high-level cognitive skills are required for collaborative hunting, Tsutsui et al. used a type of artificial intelligence known as deep reinforcement learning. This allowed them to develop a computational model in which a small number of 'agents' had the opportunity to 'learn' whether and how to work together to catch a 'prey' under various conditions. To do so, the agents were only equipped with the ability to link distinct stimuli together, such as an event and a reward; this is similar to associative learning, a cognitive process which is widespread amongst animal species.

The model showed that the challenge of capturing the prey when hunting alone, and the reward of sharing food after a successful hunt drove the agents to learn how to work together, with previous experiences shaping decisions made during subsequent hunts. Importantly, the predators started to exhibit the ability to take on distinct, complementary roles reminiscent of those observed during collaborative hunting, such as one agent chasing the prey while another ambushes it.

Overall, the work by Tsutsui et al. challenges the traditional view that only organisms equipped with high-level cognitive processes can show refined collaborative approaches to hunting, opening the possibility that these behaviors may be more widespread than originally thought – including between animals of different species.

(*Stander, 1992*; *Boesch and Boesch, 1989*). Collaborative hunts appear to be achieved through elaborate coordination with other hunters, and are often believed to be an advanced hunting strategy requiring large brains that involve high-level cognition such as aspects of theory of mind (*Boesch and Boesch-Achermann, 2000*; *Boesch, 2002*).

However, recent findings have placed this previous belief under strain. In particular, cases of intra- and inter-specific collaborative hunting have also been demonstrated in smaller-brained vertebrates such as birds (*Bednarz, 1988*), reptiles (*Dinets, 2015*), and fish (*Bshary et al., 2006*; *Steinegger et al., 2018*). It seems possible that apparently elaborate hunting behavior can emerge in a relatively simple decision process in response to ecological needs (*Steinegger et al., 2018*). However, the decision process underlying collaborative hunting remains poorly understood because most previous studies thus far have relied exclusively on behavioral observations. Observational studies are essential for documenting such natural behavior, yet it is often difficult to identify the specific decision process that results in coordinated behavior. This limitation arises because seemingly simple behavior can result from complex processes (*Evans et al., 2019*) and vice versa (*Couzin et al., 2002*).

We, therefore, sought to further our understanding of the processes underlying collaborative hunting by adopting a different approach, namely, computational multi-agent simulation based on deep reinforcement learning. Deep reinforcement learning mechanisms were originally inspired by animal associative learning (*Sutton and Barto, 1981*), and are thought to be closely related to neural mechanisms for reward-based learning centering on dopamine (*Schultz et al., 1997*; *Samejima et al., 2005*; *Doya, 2008*). Given that associative learning is likely to be the most widely adopted learning mechanism in animals (*Mackintosh, 1974*; *Wynne, 2001*), collaborative hunting could arise through associative learning, where simple decision rules are developed based on behavioral cues [i.e. contingencies of reinforcement (*Skinner, 2014*)].

Specifically, we first explored whether predator agents based on deep reinforcement learning learn decision rules resulting in collaborative hunting and, if so, under what conditions through predator-prey interactions in a computational ecological environment. We then examined what internal representations are associated with the decision rules. Furthermore, we confirmed the generality of the acquired predators' decision rules using joint plays between agents (predators) and humans (prey).

Notably, our predator agents successfully learned to collaborate in capturing their prey solely through a reinforcement learning algorithm, without employing explicit mechanisms comparable to aspects of theory of mind (*Yoshida et al., 2008*; *Foerster, 2019*; *Hu and Foerster, 2020*). Moreover, our results showed that the acquisition of decision rules resulting in collaborative hunting is facilitated by a combination of two factors: the difficulty of capturing prey during solitary hunting, and food (i.e. reward) sharing following capture. We also found that decisions underlying collaborative hunts were related to distance-dependent internal representations formed by prior experience. Furthermore, the decision rules worked robustly against unknown prey controlled by humans. These provide insight that collaborative hunts do not necessarily require sophisticated cognitive mechanisms, and simple decision rules based on mappings between states and actions can be practically useful in nature. Our results support the recent suggestions that the underlying processes facilitating collaborative hunting can be relatively simple (*Lang and Farine, 2017*).

## Results

We set out to model the decision process of predators and prey in an interactive environment. In this study, we focused on a chase and escape scenario in a two-dimensional open environment. Chase and escape is a potentially complex phenomenon in which two or more agents interact in environments that change from moment to moment. Nevertheless, many studies have shown that the rules of chase/escape behavior (e.g. which direction to move at each time in a given situation) can be described by relatively simple mathematical models consisting of the current state (e.g. positions and velocities) (*Brighton et al., 2017*; *Tsutsui et al., 2020*; *Howland, 1974*). We, therefore, considered modeling the agent's decision process in a standard reinforcement learning framework for a finite Markov decision process in which each sequence is a distinct state. In this framework, the agent interacts with the environment through a sequence of states, actions, and rewards, and aims to select actions in a manner that maximizes cumulative future reward (*Sutton and Barto, 2018*).

We modeled an agent (predator/prey) with independent learning, which is one of the simplest approaches to multi-agent reinforcement learning (*Tan, 1993*; *Figure 1a*). In this approach, each agent independently learns its own policy and treats the other agents as part of the environment. In other words, each agent learns policies that are conditioned only on its local observation history, and does not account for the non-stationarity of the multi-agent environment. That is, in contrast to previous studies on multi-agent reinforcement learning (*Yoshida et al., 2008*; *Foerster, 2019*; *Hu and Foerster, 2020*; *Tesauro, 2003*; *Foerster et al., 2016*; *Silver et al., 2017*; *Lowe, 2017*; *Foerster et al., 2018*; *Sunehag, 2017*; *Rashid, 2020*; *Son et al., 2019*; *Baker, 2019*; *Christianos et al., 2020*; *Mugan and MacIver, 2020*; *Hamrick, 2021*; *Yu, 2022*), our agents did not infer the mental states of others, did not share network parameters and value functions, and did not access models of the environment for planning. For each agent $n$, the policy $\pi^n$ was represented by a neural network and optimized using the deep Q-network framework (*Mnih et al., 2015*) (see Methods). The inputs to the neural network are the positions of a specific agent in the absolute coordinate system and the positions and velocities of a specific agent and others in the relative coordinate system with respect to the prey (or the nearest predator), which are determined based on findings in neuroscience (*O'Keefe and Dostrovsky, 1971*) and ethology (*Brighton et al., 2017*; *Tsutsui et al., 2020*), respectively. The outputs are the acceleration in 12 directions every 30° in the relative coordinate system, which is determined with reference to an ecological study (*Wilson et al., 2018*). We assumed that delays in sensorimotor processing would be compensated for by estimation of the motion of self (*Wolpert et al., 1998*; *Kawato, 1999*) and others (*Tsutsui et al., 2021*), and the current information at each time step was taken as input as is. The play area size was constrained to

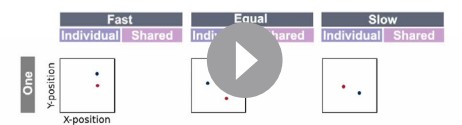

**Video 1.** Example videos in the one-predator conditions.

https://elifesciences.org/articles/85694/figures#video1

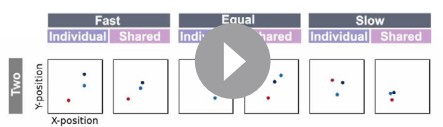

**Video 2.** Example videos in the two-predator conditions.

https://elifesciences.org/articles/85694/figures#video2

a range of –1 to 1 on the $x$ and $y$ axes, and the initial positions of the predators and prey in each episode were randomly selected from a range of –0.5 to 0.5 on the $x$ and $y$ axes. All agents (predators/prey) were represented as a disk and the diameters were set to 0.1. The predator(s) were rewarded for capturing the prey (+1), namely contacting the disks, and punished for moving out of the area (–1), and the prey was punished for being captured by the predator or for moving out of the area (–1). The time step was 0.1 and the time limit in each episode was set to 30 s. During the evaluation phase, if the predator captured the prey within the time limit, the predator was deemed successful; otherwise, the prey was considered successful. Additionally, if one side (predators/prey) moved out of the area, the other side (prey/predators) was deemed successful.

## Exploring the conditions under which collaborative hunting emerges

We first performed computational simulations with three experimental conditions to investigate the conditions under which collaborative hunting emerges (*Figure 1b*; *Videos 1–3*). As experimental conditions, we selected the number of predators, relative mobility, and prey (reward) sharing based on ecological findings (*Bailey et al., 2013*; *Lang and Farine, 2017*). For the number of predators, three conditions were set: 1 (one), 2 (two), and 3 (three). In all these conditions, the number of prey was set to 1. For the relative mobility, three conditions were set: 120% (fast), 100% (equal), and 80% (slow), which represented the acceleration of the predator, based on that of the prey. For the prey sharing, two conditions were set: with sharing (shared), in which all predators were rewarded when a predator catches the prey, and without sharing (individual), in which a predator was rewarded only when it catches the prey by itself. In total, there were 15 conditions.

As the example trajectories show, under the fast and equal conditions, the predators often caught their prey shortly after the episode began, whereas under the slow condition, the predators somewhat struggled to catch their prey (Fig. 1b). To evaluate their behavior, we calculated the proportion of predations that were successful and mean episode duration. For the fast and equal conditions, predations were successful in almost all episodes, regardless of the number of predators and the presence or absence of reward sharing (e.g. 0.99 ± 0.00 for the one × fast and one × equal conditions; *Figure 2—figure supplement 1*). This indicates that in situations where predators were faster than or equal in speed to their prey, they almost always succeeded in capturing the prey, even when they were the sole predator. Although the mean episode duration decreased with an increasing number of predators in both fast and equal conditions, the difference was small. As a whole, these results indicate that there is little benefit of cooperation among multiple predators in the fast and equal conditions. As it is unlikely that cooperation among predators will emerge under such conditions in nature from an evolutionary perspective (*Smith, 1982*; *Axelrod and Hamilton, 1981*), the analysis below is limited to the slow condition. For the slow condition, a solitary predator was rarely successful, and the proportion of predations that were successful increased with the number of predators (*Figure 2a*). Moreover, the mean duration decreased with an increasing number of predators (*Figure 2a* bottom). These results indicate that, under the slow condition, the benefits of cooperation among multiple predators are significant. In addition, except for the two × individual condition, the increase in the proportion of success with an increasing number of predators was much greater than the theoretical prediction (*Packer and Ruttan, 1988*), calculated based on the proportion of solitary hunting, assuming that each predator's performance is independent of the others' (see Methods). These results indicate that under these conditions, elaborate hunting behavior (e.g. 'collaboration') that is qualitatively different from hunting alone may emerge.

Then, we examined agent behavioral patterns and found that there were differences in the movement paths that predators take to catch their prey among the conditions (*Figure 2b*). As shown in the typical example, under the individual condition, both predators moved in a similar manner toward their prey (*Figure 2b* left) and, in contrast, under the shared condition, one predator moved toward their prey while the other predator moved along a different route (*Figure 2b* right). To ascertain their behavioral patterns, we created heat maps showing the frequency of agent presence

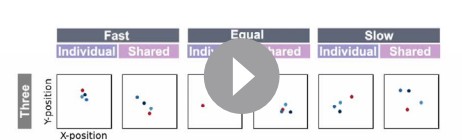

**Video 3.** Example videos in the three-predator conditions.
https://elifesciences.org/articles/85694/figures#video3

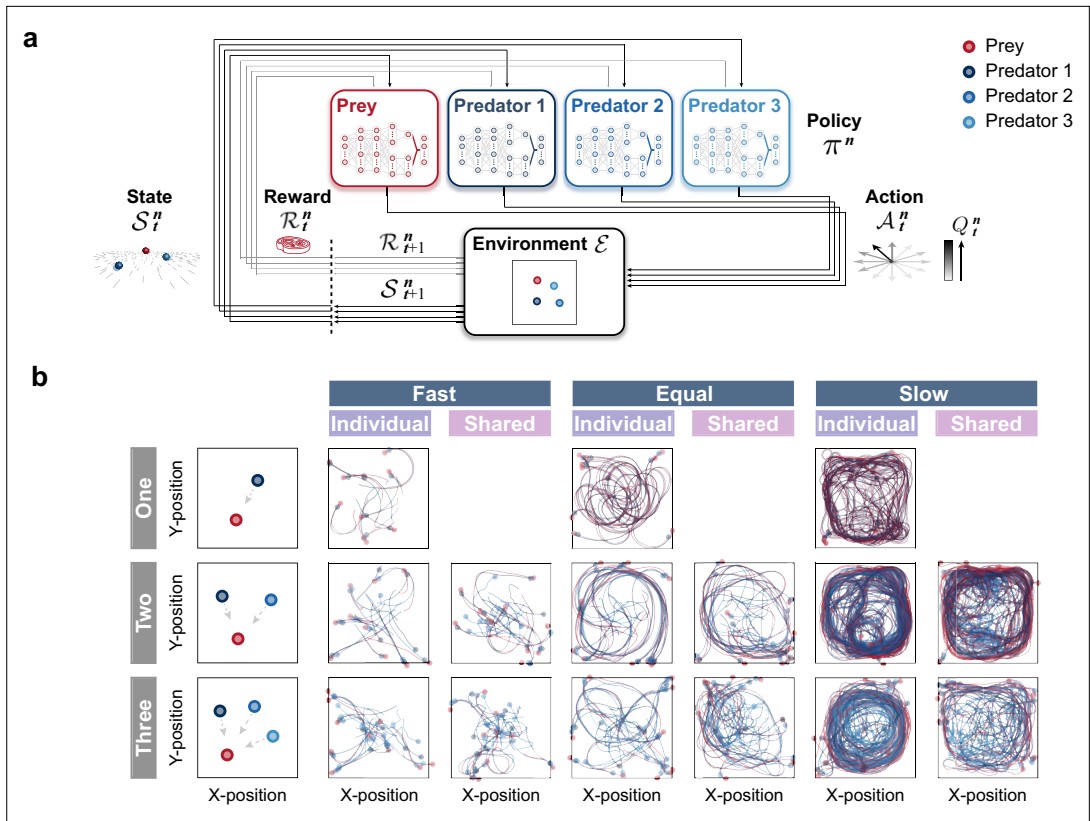

**Figure 1.** Agent architecture and examples of movement trajectories. (**a**) An agent's policy is represented by a deep neural network (see Methods). A state of the environment is given as input to the network. An action is sampled from the network's output, and the agent receives a reward and a subsequent state. The agent learns to select actions that maximize cumulative future rewards. In this study, each agent learned its policy network independently, that is, each agent treats the other agents as part of the environment. This illustration shows a case with three predators. (**b**) The movement trajectories are examples of interactions between predator(s) (dark blue, blue, and light blue) and prey (red) that overlay 10 episodes in each experimental condition. The experimental conditions were set as the number of predators (one, two, or three), relative mobility (fast, equal, or slow), and reward sharing (individual or shared), based on ecological findings.

The online version of this article includes the following figure supplement(s) for figure 1:

**Figure supplement 1.** Network architecture.

**Figure supplement 2.** Diagram of model input.

---

at each location in the area (***Figure 2c***). We found that there was a noticeable difference between the individual and shared reward conditions. In the individual condition, the heat maps of prey and respective predators were quite similar (***Figure 2c***), whereas this was not always the case in the shared condition (***Figure 2c***). In particular, the heat maps of predator 2 in the two-predator condition and predator 3 in the three-predator condition showed localized concentrations (***Figure 2c*** far right, respectively). To assess these differences among predators in more detail, we compared the predators' decisions (i.e. action selections) in these conditions with that in the one-predator condition (i.e. solitary hunts) using two indices, concordance rate, and circular correlation (***Berens, 2009***; ***Figure 2—figure supplement 2***). Following previous studies (***Scheel and Packer, 1991***), we also calculated the ratios of distance moved during hunting among predators (***Figure 2—figure supplement 3***). Overall, these findings support the idea that predators with heat maps similar to their prey acted as 'chasers' (or 'drivers'), while predators with different heat maps behaved as 'blockers' (or 'ambushers'). That is, our results show that, although most predators acted as chasers, some predators acted as blockers rather than chasers in the shared condition, indicating the emergence of collaborative hunting characterized by role divisions among predators under the condition.

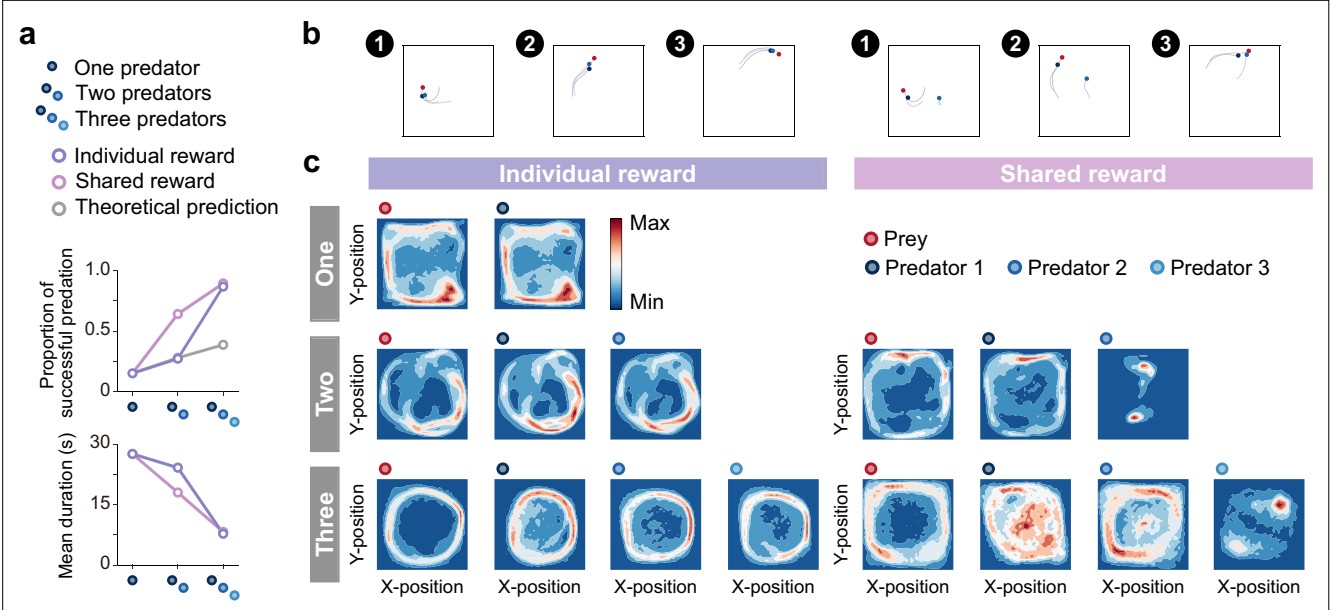

**Figure 2.** Emergence of collaborations among predators. (**a**) Proportion of predations that were successful (top) and mean episode duration (bottom). For both panels, quantitative data denote the mean of 100 episodes ± SEM across 10 random seeds. The error bars are barely visible because the variation is negligible. The theoretical prediction values were calculated based on the proportion of solitary hunts (see Methods). The proportion of predations that were successful increased as the number of predators increased ($F_{number(2,18)}$ = 1346.67, $p<0.001$; $\eta^2$ = 0.87; one vs. two: $t_{(9)}$ = 20.38, $p<0.001$; two vs. three: $t_{(9)}$ = 38.27, $p<0.001$). The mean duration decreased with increasing number of predators ($F_{number(2,18)}$ = 1564.01, $p<0.001$; $\eta^2$ = 0.94; one vs. two: $t_{(9)}$ = 15.98, $p<0.001$; two vs. three: $t_{(9)}$ = 40.65, $p<0.001$). (**b**) Typical example of different predator routes between the individual (left) and shared (right) conditions, in the two-predator condition. The numbers (1–3) show a series of state transitions (every second) starting from the same initial position. Each panel shows the agent positions and the trajectories leading up to that state. In these instances, the predators ultimately failed to capture the prey within the time limit (30 s) under the individual condition, whereas the predators successfully captured the prey in only 3 s under the shared condition. (**c**) Comparison of heat maps between individual (left) and shared (right) reward conditions. The heat maps of each agent were constructed based on the frequency of stay in each position, which was cumulative for 1000 episodes (100 episodes × 10 random seeds). In the individual condition, there were relatively high correlations between the heat maps of the prey and each predator, regardless of the number of predators (One:$r=0.95, p<0.001$, Two:$r=0.83, p<0.001$ in predator 1,$r=0.78, p<0.001$ in predator 2, Three:$r=0.41, p<0.001$ in predator 1,$r=0.56, p<0.001$ in predator 2,$r=0.45, p<0.001$ in predator 3). In contrast, in the shared condition, only one predator had a relatively high correlation, whereas the others had low correlations (Two:$r=0.65, p<0.001$ in predator 1,$r=0.01, p=0.80$ in predator 2, Three:$r=0.17, p<0.001$ in predator 1,$r=0.54, p<0.001$ in predator 2,$r=0.03, p=0.23$ in predator 3).

The online version of this article includes the following figure supplement(s) for figure 2:

**Figure supplement 1.** Proportion of predations that were successful, mean episode duration, and heat maps for each condition.

**Figure supplement 2.** Circular histogram, concordance rate, and circular correlation.

**Figure supplement 3.** Scaled distance among predators and proportion of prey capture.

**Figure supplement 4.** Typical example of coordinated hunting behavior in the three × individual condition.

## Mechanistic interpretability of collaboration

We next sought the predators' internal representations to better understand how such collaborative hunting is accomplished. Using a two-dimensional t-distributed stochastic neighbor embedding (t-SNE) (*van der Maaten and Hinton, 2008*), we visualized the last hidden layers of the state and action streams in the policy network as internal representations of agents (*Figure 3*, *Figure 3—figure supplements 1–3*). To understand how each agent represents its environment and what aspects of the state are well represented, we examined the relationship between the scenes of a typical scenario and their corresponding points on the embedding (*Figure 3a and b*). As expected, when the predator is likely to catch its prey (e.g. scene 4), the predator estimated a higher state value, whereas, when the predator is not (e.g. scene 5), the predator estimated a lower state value (*Figure 3a* top). Related to this, the variance of action values tends to be larger for both predator and prey when they are close (*Figure 3a* bottom), indicating that the difference in the value of choosing each action is greater when the choice of action is directly related to the reward (see also *Figure 3—figure supplement 4*). These



**Figure 3.** Embedding of internal representations underlying collaborative hunting. (**a**) Two-dimensional t-distributed stochastic neighbor embedding (t-SNE) embedding of the representations in the last hidden layers of the state-value stream (top) and action-value stream (bottom) in the shared reward condition. The representation is assigned by the policy network of each agent to states experienced during predator-prey interactions. The points are colored according to the state values and standard deviation of the action values, respectively, predicted by the policy network (ranging from dark red (high) to dark blue (low)). (**b**) Corresponding states for each number in each embedding. The number (1–5) in each embedding corresponds to a selected series of state transitions. The series of agent positions in the state transitions (every second) and, for ease of visibility, the trajectories leading up to that state are shown. (**c**) Embedding colored according to the distances between predators and prey in the individual (left) and shared (right) reward conditions. Distances 1 and 2 denote the distances between predator 1 and prey and predator 2 and prey, respectively. If both distances are short, the point is colored blue; if both are long, it is colored white.

The online version of this article includes the following figure supplement(s) for figure 3:

**Figure supplement 1.** Two-dimensional t-distributed stochastic neighbor embedding (t-SNE) embedding of the representations in the last hidden layers of the state-value stream (top) and action-value stream (bottom) in the individual reward condition, in the slow × two conditions.

**Figure supplement 2.** Two-dimensional t-distributed stochastic neighbor embedding (t-SNE) embedding colored according to the absolute coordinates of itself in the individual (left) and shared (right) reward conditions, in the slow × two conditions.

**Figure supplement 3.** Two-dimensional t-distributed stochastic neighbor embedding (t-SNE) embedding of the representations in the last hidden layers ofstate-value stream and action-value stream, in the slow × three conditions.

**Figure supplement 4.** Corresponding state-action values (Q-values) for each state.

**Figure supplement 5.** Rule-based predator agent architectures.

**Figure supplement 6.** Movement trajectories (left) and heat maps (right) of the rule-based predator agents.

**Figure supplement 7.** Two-dimensional t-distributed stochastic neighbor embedding (t-SNE) embedding of the representations in the last hidden layers ofthe linear network (top) and the nonlinear network (bottom) in behavioral cloning.

**Figure supplement 8.** Histogram of the state value (V-value) in the individual (left) and shared (right) conditions.

**Figure supplement 9.** Histogram of the standard deviation of state-action values (Q-values) in individual (left) and shared (right) conditions.

*Figure 3 continued on next page*

*Figure 3 continued*

**Figure supplement 10.** Histogram of the distance between the prey and each predator in individual (left) and shared (right) conditions.

**Figure supplement 11.** Histogram of the distance between the prey and each predator in the simulations, using rule-based predator agents.

results suggest that the agents were able to learn the networks that output the estimations of state and action values consistent with our intuition.

Furthermore, we found a distinct feature in the embedding of predators' representations. Specifically, in certain state transitions, the position of the points on the embedding changed little, even though the agents were moving (e.g. scenes 1–2 on the embedding of the predator 2). From this, we deduced that the predators' representations may be more focused on encoding the distance between themselves and others, rather than the specific locations of both parties. To test our reasoning, we colored the representations according to the distance between predators and prey; distance 1 denotes the distance between predator 1 and the prey, and distance 2 denotes that between predator 2 and the prey. As a result, the representations of predators in the shared condition could be clearly separated by the distance-dependent coloration (Fig. 3c right), in contrast to those in the individual condition (Fig. 3c left). These indicate that the predators in the shared condition estimated state and action values and made decisions associated with distance-dependent representations (see *Figure 3—figure supplement 2* for the prey's decision).

## Evaluating the playing strength of predator agents using joint play with humans

Finally, to verify the generality of predators' decisions against unknown prey, we conducted an experiment of joint play between agents and humans. In the joint play, human participants controlled prey on a screen using a joystick. The objective, as in the computational simulation described above, was to evade capture until the end of the episode (30 s) while remaining within the area. We found that

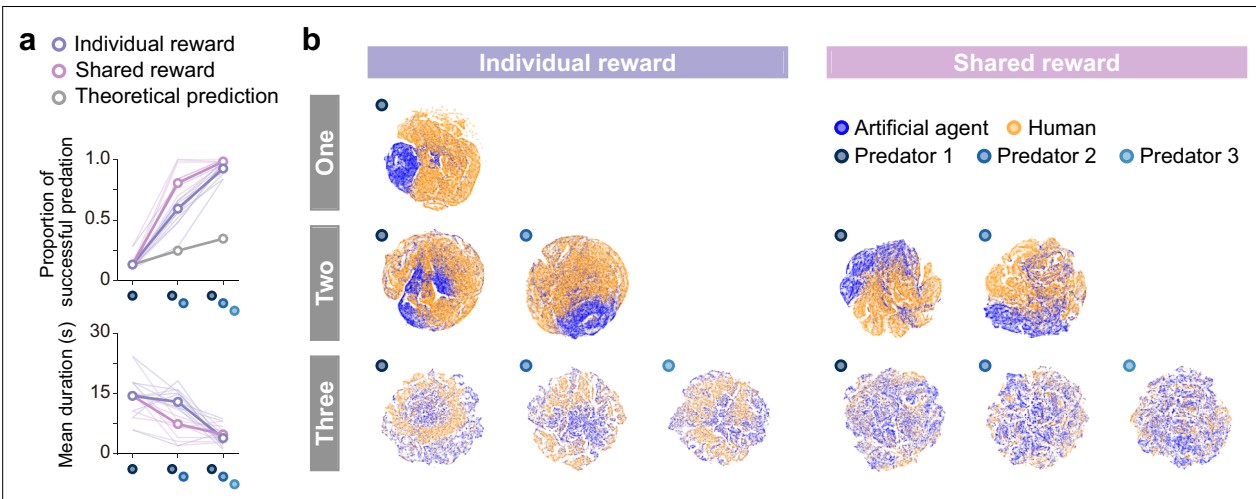

**Figure 4.** Superior performance of predator agents for prey controlled by humans and comparison of internal representations. (**a**) Proportion of predations that were successful (top) and mean episode duration (bottom). For both panels, the thin line denotes the performance of each participant, and the thick line denotes the mean. The theoretical prediction values were calculated based on the mean of proportion of solitary hunts. The proportion of predations that were successful increased as the number of predators increased ($F_{\text{number}(1.28, 11.48)}$ = 276.20, $p<0.001$; $\eta^2$ = 0.90; one vs. two: $t_{(9)}$ = 13.80, $p<0.001$; two vs. three: $t_{(9)}$ = 5.9402, $p<0.001$). The mean duration decreased with an increasing number of predators ($F_{\text{number}(2, 18)}$ = 23.77, $p<0.001$; $\eta^2$ = 0.49; one vs. two: $t_{(9)}$ = 2.60, $p=0.029$; two vs. three: $t_{(9)}$ = 5.44, $p<0.001$). (**b**) Comparison of two-dimensional t-distributed stochastic neighbor embedding (t-SNE) embedding of the representations in the last hidden layers of state-value stream between self-play (predator agents vs. prey agent) and joint play (predator agents vs. prey human).

The online version of this article includes the following figure supplement(s) for figure 4:

**Figure supplement 1.** Comparison of two-dimensional t-distributed stochastic neighbor embedding (t-SNE) embedding of the internal representations.

**Figure supplement 2.** Comparison of heat maps between individual (left) and shared (right) reward conditions in joint play.

the outcomes of the joint play showed similar trends to those of the computer simulation (*Figure 4a*), showing that the proportion of predations that were successful increased and the mean episode duration decreased as the number of predators increased. These indicate that the predator agents' decision rules worked well for the prey controlled by humans. To visualize the associations of states experienced by predator agents versus agents and versus humans, we show colored two-dimensional t-SNE embedding of the representations in the last hidden layers of the state and action streams (*Figure 4b*, *Figure 4—figure supplement 1*). These showed that, in contrast to a previous study (*Mnih et al., 2015*), the states were quite distinct, suggesting that predator agents experienced unfamiliar states when playing against the prey controlled by humans. This unfamiliarity may make it difficult for predators to make proper decisions. Indeed, in the one-predator condition, the predator agent occasionally exhibited odd behavior (e.g. staying in one place; see *Figure 4—figure supplement 2*). On the other hand, in the two- and three-predator conditions, predator agents rarely exhibited such behavior and showed superior performance. This indicates that decision rules of cooperative hunting acquired in certain environments could be applied in other somewhat different environments.

## Discussion

Collaborative hunting has been traditionally thought of as an advanced hunting strategy that involves high-level cognition such as aspects of theory of mind (*Boesch and Boesch-Achermann, 2000*; *Boesch, 2002*). Here, we have shown that 'collaboration'(*Boesch and Boesch, 1989*) can emerge in group hunts of artificial agents based on deep reinforcement learning. Notably, our predator agents successfully learned to collaborate in capturing their prey solely through a reinforcement learning algorithm, without employing explicit mechanisms comparable to aspects of theory of mind (*Yoshida et al., 2008*; *Foerster, 2019*; *Hu and Foerster, 2020*). This means that, in contrast to the traditional view, apparently elaborate coordination can be accomplished by relatively simple decision rules, that is, mappings between states and actions. This result advances our understanding of cooperative hunting behavior and its decision process, and may offer a novel perspective on the evolution of sociality.

Our results on agent behavior are broadly consistent with previous studies concerning observations of animal behavior in nature. First, as the number of predators increased, success rates increased and hunting duration decreased (*Creel and Creel, 1995*). Second, whether collaborative hunts emerge depended on two factors: the success rate of hunting alone (*Busse, 1978*; *Boesch, 2002*) and the presence or absence of reward sharing following prey capture (*Boesch, 1994*; *Stanford, 1996*). Third, while each predator generally maintained a consistent role during repeated collaborative hunts, there was flexibility for these roles to be swapped as needed (*Stander, 1992*; *Boesch, 2002*). Finally, predator agents in this study acquired different strategies depending on the conditions despite having exactly the same initial values (i.e. network weights), resonating with the findings that lions and chimpanzees living in different regions exhibit different hunting strategies (*Stander, 1992*; *Boesch-Achermann and Boesch, 1994*). These results suggest the validity of our computational simulations and highlight the close link between predators' behavioral strategies and their living environments, such as the presence of other predators and sharing of prey.

The collaborative hunts have shown performance that surpasses the theoretical predictions based on solitary hunting outcomes. This result is in line with the notion that role division among predators in nature could provide fitness benefits (*Lang and Farine, 2017*; *Boesch and Boesch-Achermann, 2000*). Meanwhile, when three predators were involved, performance was comparable whether prey was shared or not. One possible factor that has caused this is spatial constraints. We found that predators occasionally block the prey's escape path, exploiting the boundaries of the play area and the chasing movements of other predators even in the individual reward condition (*Figure 2—figure supplement 4*). These results suggest that, under certain scenarios, coordinated hunting behaviors that enhance the success rate of predators may emerge regardless of whether food is shared, potentially relating to the benefits of social predation, including interspecific hunting (*Bshary et al., 2006*; *Thiebault et al., 2016*; *Sampaio et al., 2021*).

We found that the mappings resulting in collaborative hunting were related to distance-dependent internal representations. Additionally, we showed that the distance-dependent rule-based predators successfully reproduced behaviors similar to those of the deep reinforcement learning predators, supporting the association between decisions and distances (Methods; *Figure 3—figure supplement*

*5*, *Figure 3—figure supplement 6* and *Figure 3—figure supplement 7*). Deep reinforcement learning has held the promise for providing a comprehensive framework for studying the interplay among learning, representation, and decision making (*Botvinick et al., 2020*; *Mobbs et al., 2021*), but such efforts for natural behavior have been limited (*Banino et al., 2018*; *Jaderberg et al., 2019*). Our result that the distance-dependent representations relate to collaborative hunting is reminiscent of a recent idea about the decision rules obtained by observation in fish (*Steinegger et al., 2018*). Notably, the input variables of predator agents do not include variables corresponding to the distance(s) between the other predator(s) and prey, and this means that the predators in the shared conditions acquired the internal representation relating to distance to prey, which would be a geometrically reasonable indicator, by optimization through interaction with their environment. Our results suggest that deep reinforcement learning methods can extract systems of rules that allow for the emergence of complex behaviors.

The predator agents' decision rules (i.e. policy networks) acquired through interactions with other agents (i.e. self-play) were also useful for unknown prey controlled by humans, despite the dissociation of the experienced states. This suggests that decision rules formed by associative learning can successfully address natural problems, such as catching prey with somewhat different movement patterns than one's usual prey. Note that the learning mechanism of associative learning (or reinforcement learning) is relatively simple, but it allows for flexible behavior in response to situations, in contrast to innate and simple stimulus-response. Indeed, our prey agents achieved a higher rate of successful evasions than those operated by humans. Our view that decisions for successful hunting are made through representations formed by prior experience is a counterpart to the recent idea that computational relevance for successful escape may be cached and ready to use, instead of being computed from scratch on the spot (*Evans et al., 2019*). If animals' decision processes in predator-prey dynamics are structured in this way, it could be a product of natural selection, enabling rapid, robust, and flexible action in interactions with severe time constraints.

In conclusion, we demonstrated that the decisions underlying collaborative hunting among artificial agents can be achieved through mappings between states and actions. This means that collaborative hunting can emerge in the absence of explicit mechanisms comparable to aspects of theory of mind, supporting the recent idea that collaborative hunting does not necessarily rely on complex cognitive processes in brains (*Lang and Farine, 2017*). Our computational ecology is an abstraction of a real predator-prey environment. Given that chase and escape often involve various factors, such as energy cost (*Hubel et al., 2016*), partial observability (*Mugan and MacIver, 2020*; *Hunt et al., 2021*), signal communication (*Vail et al., 2013*), and local surroundings (*Evans et al., 2019*), these results are only a first step on the path to understanding real decisions in predator-prey dynamics. Furthermore, exploring how mechanisms comparable to aspects of theory of mind (*Yoshida et al., 2008*; *Foerster, 2019*; *Hu and Foerster, 2020*) or the shared value functions (*Lowe, 2017*; *Foerster et al., 2018*; *Rashid, 2020*), which are increasingly common in multi-agent reinforcement learning, play a role in these interactions could be an intriguing direction for future research. We believe that our results provide a useful advance toward understanding natural value-based decisions and forge a critical link between ecology, ethology, psychology, neuroscience, and computer science.

## Methods

### Environment

The predator and prey interacted in a two-dimensional world with continuous space and discrete time. This environment was constructed by modifying an environment known as 'predator-prey' within a multi-agent particle environment (*Lowe, 2017*). Specifically, the position of each agent was calculated by integrating the acceleration (i.e. selected action) twice with the Euler method, and viscous resistance proportional to velocity was considered. The modifications were that the action space (play area size) was constrained to the range of –1 to 1 on the $x$ and $y$ axes, all agent (predator/prey) disk diameters were set to 0.1, landmarks (obstacles) were eliminated, and predator-to-predator contact was ignored for simplicity (*Tsutsui et al., 2022*). The predator(s) was rewarded for capturing the prey (+1), namely contacting the disks, and punished for moving out of the area (–1), and the prey was penalized for being captured by the predator or for moving out of the area (–1). The predator and prey were represented as a red and blue disk, respectively, and the play area was represented as a black square

enclosing them. The time step was 0.1 s and the time limit in each episode was set to 30 s. The initial positions of the predators and prey in each episode were randomly selected from a range of –0.5 to 0.5 on the *x* and *y* axes.

## Experimental conditions

We selected the number of predators, relative mobility, and prey (reward) sharing as experimental conditions, based on ecological findings (*Bailey et al., 2013*; *Lang and Farine, 2017*). For the number of predators, three conditions were set: 1 (one), 2 (two), and 3 (three). In all these conditions, the number of prey was set to 1. For the relative mobility, three conditions were set: 120% (fast), 100% (equal), and 80% (slow) for the acceleration exerted by the predator, based on that exerted by the prey. For the prey sharing, two conditions were set: with sharing (shared), in which all predators were rewarded when a predator catches the prey, and without sharing (individual), in which a predator was rewarded only when it catches prey by itself. In total, there were 15 conditions.

## Agent architecture

We considered a sequential decision-making setting in which a single agent interacts with an environment $\mathcal{E}$ in a sequence of observations, actions, and rewards. At each time-step $t$, the agent observes a state $s_t \in \mathcal{S}$ and selects an action $a_t$ from a discrete set of actions $\mathcal{A} = \{1, 2, \ldots, |\mathcal{A}|\}$. One time step later, in part as a consequence of its action, the agent receives a reward, $r_{t+1} \in \mathcal{R}$, and moves itself to a new state $s_{t+1}$. In the MDP, the agent thereby gives rise to a sequence that begins as follows: $s_0, a_0, r_1, s_1, a_1, r_2, s_2, a_2, r_3, \ldots$, and learns a behavioral rule (policy) that depends upon these sequences.

The goal of the agent is to maximize the expected discounted return over time through its choice of actions (*Sutton and Barto, 2018*). The discounted return $R_t$ was defined as $\sum_{k=0}^{T} \gamma^k r_{t+k+1}$, where $\gamma \in [0, 1]$ is a parameter called the discount rate that determines the present value of future rewards, and $T$ is the time step at which the task terminates. The state-value function, action-value function, and advantage function are defined as $V^\pi(s) = \mathbb{E}_\pi[R_t|s_t = s]$, $Q^\pi(s, a) = \mathbb{E}_\pi[R_t|s_t = s, a_t = a]$, and $A^\pi(s, a) = Q^\pi(s, a) - V^\pi(s)$, respectively, where $\pi$ is a policy mapping states to actions. The optimal action-value function $Q^\star(s, a)$ is then defined as the maximum expected discounted return achievable by following any strategy, after observing some state $s$ and then taking some action $a$, $Q^\star(s, a) = \max_\pi \mathbb{E}[R_t|s_t = s, a_t = a, \pi]$. The optimal action-value function can be computed by finding a fixed point of the Bellman equations:

$$Q^\star(s, a) = \mathbb{E}_{s' \sim \mathcal{E}}\left[r + \gamma \max_{a'} Q^\star(s', a'|s, a)\right], \tag{1}$$

where $s'$ and $a'$ are the state and action at the next time-step, respectively. This is based on the following intuition: if the optimal value $Q^\star(s', a')$ of the state $s'$ was known for all possible actions $a'$, the optimal strategy is to select the action $a'$ maximizing the expected value of $r + \gamma \max_{a'} Q^\star(s', a')$. The basic idea behind many reinforcement learning algorithms is to estimate the action-value function by using the Bellman equation as an iterative update; $Q_{i+1}(s, a) = \mathbb{E}[r + \gamma \max_{a'} Q_i(s', a'|s, a)]$. Such value iteration algorithms converge to the optimal action-value function in situations where all states can be sufficiently sampled, $Q_i \to Q^\star$ as $i \to \infty$. In practice, however, it is often difficult to apply this basic approach, which estimates the action-value function separately for each state, to real-world problems. Instead, it is common to use a function approximator to estimate the action-value function, $Q(s, a; \theta) \approx Q^\star(s, a)$.

There are several possible methods for function approximation, yet we here use a neural network function approximator referred to as deep *Q*-network (DQN) (*Mnih et al., 2015*) and some of its extensions to overcome the limitations of the DQN, namely Double DQN (*Van Hasselt et al., 2016*), Prioritized Experience Replay (*Schaul et al., 2015*), and Dueling Networks (*Wang, 2016*). Naively, a *Q*-network with weights $\theta$ can be trained by minimizing a loss function $\mathcal{L}(\theta)$ that changes at each iteration $i$,

$$\mathcal{L}_i(\theta_i) = \mathbb{E}_{s, a \sim \rho(\cdot)}\left[\frac{1}{2}\left(y_i - Q(s, a; \theta_i)\right)^2\right], \tag{2}$$

where $y_i = r + \gamma \max_{a'} Q(s', a'; \theta_{i-1}|s, a)$ is the target value for iteration $i$, and $\rho(s, a)$ is a probability distribution over states $s$ and actions $a$. The parameters from the previous iteration $\theta_{i-1}$ are kept constant when optimizing the loss function $\mathcal{L}(\theta)$. By differentiating the loss function with respect to the weights we arrive at the following gradient,

$$\nabla_{\theta_i} \mathcal{L}_i(\theta_i) = \mathbb{E}_{s,a \sim \rho(\cdot); s' \sim \mathcal{E}}\left[\left(r + \gamma \max_{a'} Q(s', a'; \theta_{i-1}) - Q(s, a; \theta_i)\right)\nabla_{\theta_i} Q(s, a; \theta_i)\right]. \tag{3}$$

We could attempt to use the simplest $Q$-learning to learn the weights of the network $Q(s, a; \theta)$ online; however, this estimator performs poorly in practice. In this simplest form, they discard incoming data immediately, after a single update. This results in two issues: (i) strongly correlated updates that break the i.i.d. assumption of many popular stochastic gradient-based algorithms and (ii) the rapid forgetting of possibly rare experiences that would be useful later. To address both of these issues, a technique called experience replay is often adopted (**Lin, 1992**), in which the agent's experiences at each time-step $e_t = (s_t, a_t, r_{t+1}, s_{t+1})$ are stored in a dataset (also referred to as replay memory) $\mathcal{D} = \{e_1, e_2, \ldots, e_N\}$, where $N$ is the dataset size, for some time period. When training the $Q$-network, instead of only using the current experience as prescribed by standard $Q$-learning, mini-batches of experiences are sampled from $\mathcal{D}$ uniformly, at random, to train the network. This enables breaking the temporal correlations by mixing more and fewer recent experiences for the updates, and rare experiences will be used for more than just a single update. Another technique, called the target-network, is also often used for updating to stabilize learning. To achieve this, the target value $y_i$ is replaced by $r + \gamma \max_{a'} Q(s', a'; \theta_i^-)$, where $\theta_i^-$ are the weights, which are frozen for a fixed number of iterations. The full algorithm combining these ingredients, namely experience replay and the target-network, is often called a deep Q-network (DQN), and its loss function takes the form:

$$\mathcal{L}_i(\theta_i) = \mathbb{E}_{(s,a,r',s') \sim \mathcal{U}(\mathcal{D})}[(y_i^{DQN} - Q(s, a; \theta_i))^2], \tag{4}$$

where

$$y_i^{DQN} = r + \gamma \max_{a'} Q(s', a'; \theta_i^-), \tag{5}$$

and $\mathcal{U}(\cdot)$ is a uniform sampling.

It has become known that $Q$-learning algorithms perform poorly in some stochastic environments. This poor performance is caused by large overestimations of action values. These overestimations result from a positive bias that is introduced because $Q$-learning uses the maximum action value as an approximation for the maximum expected action value. As a method to alleviate the performance degradation due to the overestimation, Double $Q$-learning, which decomposes the maximum operation into action selection and action evaluation by introducing the double estimator, was proposed (**Hasselt, 2010**). Double DQN (DDQN) is an algorithm that applies the Double $Q$-learning method to DQN (**Van Hasselt et al., 2016**). For the DDQN, in contrast to the original Double $Q$-learning and the other proposed method (**Fujimoto et al., 2018**), the target network in the DQN architecture, although not fully decoupled, was used as the second value function, and the target value in the loss function (i.e. Eq. Agent architecture) for iteration $i$ is replaced as follows:

$$y_i^{DDQN} = r + \gamma Q\left(s', \arg\max_{a'} Q(s', a'; \theta_i); \theta^-\right). \tag{6}$$

Prioritized Experience Replay is a method that aims to make the learning more efficient and effective than if all transitions were replayed uniformly (**Schaul et al., 2015**). For the prioritized replay, the probability of sampling from the data-set for transition $i$ is defined as

$$P(i) = \frac{p_i^\alpha}{\sum_k p_k^\alpha}, \tag{7}$$

where $p_i > 0$ is the priority of transition for iteration $i$ and the exponent $\alpha$ determines how much prioritization is used, with $\alpha = 0$ corresponding to uniform sampling. The priority $p_i$ is determined by $p_i = |\delta_i| + \epsilon$, where $\delta_i$ is a temporal-difference (TD) error (e.g. $\delta_i = r + \gamma \max_{a'} Q(s', a'; \theta_i^-) - Q(s, a; \theta_i)$ in DQN) and $\epsilon$ is a small positive constant that prevents the case of transitions not being revisited once

their error is zero. Prioritized replay introduces sampling bias, and therefore changes the solution to which the estimates will converge. This bias can be corrected by importance-sampling (IS) weights $w_i = \left( \frac{1}{N} \frac{1}{P(i)} \right)^{\beta}$ that fully compensate for the non-uniform probabilities $P(i)$ if $\beta = 1$.

Dueling Network is a neural network architecture designed for value-based algorithms such as DQN (*Wang, 2016*). This features two streams of computation, the value and advantage streams, sharing a common encoder, and is merged by an aggregation module that produces an estimate of the state-action value function. Intuitively, we can expect the dueling network to learn which states are (or are not) valuable, without having to learn the effect of each action for each state. For the reason of stability of the optimization, the last module of the network is implemented as follows:

$$Q(s, a; \theta, \eta, \xi) = V(s; \theta, \xi) + \left( A(s, a; \theta, \eta) - \frac{1}{|\mathcal{A}|} \sum_{a'} A(s, a'; \theta, \eta) \right), \tag{8}$$

where $\theta$ denotes the parameters of the common layers, whereas $\eta$ and $\xi$ are the parameters of the layers of the two streams, respectively.

We here modeled an agent (predator/prey) with independent learning, one of the simplest approaches to multi-agent reinforcement learning (*Tan, 1993*). In this approach, each agent independently learns its own policy and treats the other agents as part of the environment. In other words, each agent learns policies that are conditioned only on their local observation history, and do not account for the non-stationarity of the multi-agent environment. That is, in contrast to previous studies on multi-agent reinforcement learning (*Tesauro, 2003*; *Foerster et al., 2016*; *Silver et al., 2017*; *Lowe, 2017*; *Foerster et al., 2018*; *Sunehag, 2017*; *Rashid, 2020*; *Son et al., 2019*; *Baker, 2019*; *Christianos et al., 2020*; *Mugan and MacIver, 2020*; *Hamrick, 2021*; *Yu, 2022*), our agents did not share network parameters and value functions, and did not access models of the environment for planning. For each agent $n$, the policy $\pi^n$ is represented by a neural network and optimized, with the framework of DQN including DDQN, Prioritized Experience Replay, and Dueling architecture. The loss function of each agent takes the form:

$$\mathcal{L}_i(\theta_i, \eta_i, \xi_i) = \mathbb{E}_{s,a,r',s' \sim \mathcal{P}(\mathcal{D})} \left[ \left( y_i - Q(s, a; \theta_i, \eta_i, \xi_i) \right)^2 \right], \tag{9}$$

where

$$y_i = r + \gamma Q \left( s', \arg\max_{a'} Q(s', a'; \theta_i, \eta_i, \xi_i); \theta_i^-, \eta_i^-, \xi_i^- \right), \tag{10}$$

and $\mathcal{P}(\cdot)$ is a prioritized sampling. For simplicity, we omitted the agent index $n$ in these equations.

## Training details

The neural network was composed of four layers (*Figure 1—figure supplement 1*). There was a separate output unit for each possible action, and only the state representation was an input to the neural network. The inputs to the neural network were the positions of a specific agent in the absolute coordinate system ($x$- and $y$-positions) and the positions and velocities of a specific agent and others in the relative coordinate system ($u$- and $v$-positions and $u$- and $v$-velocities) (*Figure 1—figure supplement 2*), which were determined based on findings in neuroscience (*O'Keefe and Dostrovsky, 1971*) and ethology (*Brighton et al., 2017*; *Tsutsui et al., 2020*), respectively. We assumed that delays in sensory processing were compensated for by estimation of motion of self (*Wolpert et al., 1998*; *Kawato, 1999*) and others (*Tsutsui et al., 2021*), and the current information at each time was used as input as is. The outputs were the acceleration in 12 directions every $30°$ in the relative coordinate system, which were determined with reference to an ecological study (*Wilson et al., 2018*). After the first two hidden layers of the MLP with 64 units, the network branched off into two streams. Each branch had one MLP layer with 32 hidden units. ReLU was used as the activation function for each layer (*Glorot et al., 2011*). The network parameters $\theta^n$, $\eta^n$, and $\xi^n$ were iteratively optimized via stochastic gradient descent with the Adam optimizer (*Kingma and Ba, 2014*). In the computation of the loss, we used Huber loss to prevent extreme gradient updates (*Huber, 1992*). The model was trained for $10^6$ episodes, and the network parameters were copied to the target-network every 2000 episodes. The replay memory size was $10^4$, the minibatch size during training was 32, and the learning rate was $10^{-6}$.

The discount factor $\gamma$ was set to 0.9, and $\alpha$ was set to 0.6. We used an $\varepsilon$-greedy policy as the behavior policy $\pi^n$, which chooses a random action with probability $\varepsilon$ or an action according to the optimal $Q$ function $\arg\max_{a \in \mathcal{A}} Q^\star(s, a)$ with probability $1 - \varepsilon$. In this study, $\varepsilon$ was annealed linearly from 1 to 0.1 over the first $10^4$ episodes and fixed at 0.1 thereafter.

## Evaluation

The model performance was evaluated using the trained model. The initial position of each agent and termination criteria in each episode were the same as in training. During the evaluation, $\varepsilon$ was set to 0, and each agent took greedy actions. If the predator captured the prey within the time limit, the predator was deemed successful; otherwise, the prey was considered successful. Additionally, if one side (predators/prey) moved out of the area, the other side (prey/predators) was deemed successful. We first conducted a computational experiment (self-play: predator agent vs. prey agent). and then conducted a human behavioral experiment (joint play: predator agent vs. prey human). In the computational experiment, we simulated 100 episodes for each of the 10 random seeds (i.e. different initial positions), for a total of 1000 episodes in each condition. In the joint play, human participants controlled prey on a screen using a joystick and interacted with the predator agents for 50 episodes in each condition.

## Participants

Ten males participated in the experiment (aged 22–25, mean = 23.5, s.d.=1.2). All participants were right-handed but one, had normal or corrected-to-normal vision, and were naïve to the purpose of the study. This study was approved by the Ethics Committee of Nagoya University Graduate School of Informatics (No. 2021–27). Informed consent was obtained from each participant before the experiment. Participants received 1000 yen per hour as a reward.

## Apparatus

Participants were seated in a chair, and they operated the joystick of an Xbox One controller that could tilt freely in any direction to control a disk on the screen. The stimuli were presented on a 26.5-inch monitor (EIZO EV2730Q) at a refresh rate of 60 Hz. A gray square surrounding the disks was defined as the play area. The diameter of each disk on the screen was 2.0 cm, and the width and height of the area were 40.0 cm. The acceleration of each disk on the screen was determined by the inclination of the joystick. Specifically, acceleration was added when the degree of joystick tilt exceeded half of the maximum tilt, and the direction of the acceleration was selected from 12 directions, discretized every 30 degrees in an absolute coordinate system corresponding to the direction of joystick tilt. The reason for setting the direction of acceleration with respect to the absolute coordinate system, rather than the relative coordinate system, in the human behavioral experiment was to allow participants to control more intuitively. The position and velocity of each disk on the screen were updated at 10 Hz (corresponding to the computational simulation) and the position during the episodes was recorded at 10 Hz on a computer (MacBook Pro) with Psychopy version 3.0. The viewing distance of the participants was about 60 cm.

## Design

Participants controlled a red disk representing the prey on the screen. They were asked to evade the predator for 30 s without leaving the play area. The agent's initial position and the outcome of the episode were determined as described above. The experimental block consisted of five sets of 10 episodes, with a warm-up of 10 episodes so that participants could become accustomed to the task. In this experiment, we focused on the slow condition and there were thus five experimental conditions (one, two × individual, two × shared, three × individual, and three × shared). Each participant played one block (i.e. 50 episodes) of each experimental condition. The order of the experimental conditions was pseudo-randomized across participants.

## Rule-based agent

We constructed rule-based predator agents to test whether they could reproduce similar behavior to predator agents based on deep reinforcement learning in the two × shared condition. For consistency with the deep reinforcement learning agents, the input to the rule-based agent used to make

decisions was limited to the current information (e.g. position and velocity) and the output was provided in a relative coordinate system to the prey; that is, action 1 denotes movement toward the prey and action 7 denotes movement in the opposite direction of the prey. The predator agent first determines whether it, or another predator, is closer to the prey, and then, if the other predator is closer, it determines whether the distance 2 is less than a certain distance threshold (set to 0.4 in our simulation). The decision rule for each predator is selected by this branching, with predator 1 adopting the three rules 'chase,' 'shortcut,' and 'approach,' and predator 2 adopting the two rules 'chase' and 'ambush.' For the chase, the predator first determines whether it is near the outer edge of the play area and, if so, selects actions that will prevent it from leaving the play area. Specifically, if the predator's position is such that $|x| > 0.9$ and $|y| > 0.9$, action 3 for clockwise (CW) and action 11 for counterclockwise (CCW) was selected, respectively, and if $0.8 < |x| \leq 0.9$ and $0.8 < |y| \leq 0.9$, action 2 for CW and action 12 for CCW was selected. The CW and CCW were determined by the absolute position of the prey and the relative position vector between the closer predator and prey; the play area was divided into four parts based on the signs of the $x$ and $y$ coordinates, and CW and CCW were determined by the correspondence between each area and the sign of the larger component of absolute value ($x$ or $y$) of the relative position vector. For instance, if the closer predator is at (0.2, 0.3) and the prey is at (0.5, 0.2), it is determined to be CW. If the predator is not outside the play area, then it determines whether the prey is inside the play area, and, if so, selects actions that will drive them outside; if the prey's position is such that $|x| \leq 0.5$ and $|y| \leq 0.5$, action 11 for CW and action 3 for CCW was selected, and if $0.5 < |x| \leq 0.6$ and $0.5 < |y| \leq 0.6$, action 12 for CW and action 2 for CCW was selected. In other situations, the predator selects actions so that the direction of movement is aligned with that of the prey; if the angle of the velocity vectors between the predator and prey $\psi \leq -50$ action 3, and if $-50 < \psi \leq -15$ action 2, if $-15 < \psi \leq 15$ action 1, if $15 < \psi \leq 50$ action 2, if $50 < \psi$ action 3 was selected. For the shortcut, the predator determines whether it is near the outer edge of the play area, and if so, selected the action described above, otherwise, it selected actions that producing shorter paths to the prey; action 2 for CW and action 12 for CCW was selected. For the approach, the predator determines whether it is near the outer edge of the play area, and if so, selected the action described above, otherwise, it selected actions that move it toward the prey; action 1 was selected. For the ambush, the predator selected actions that move it toward the top center or bottom center of the play area and to remain that location until the situation changes. If the predator's position is such that $|y| \leq 0$, the predator moved with respect to the bottom center point (–0.1, 0.5), and if $|y| > 0$, it moved toward the top center point (0, 0.6). The coordinates of the top center and bottom center points were based on the result of deep reinforcement learning agents. Specifically, we first divided the play area into four parts based on the signs of the $x$ and $y$ coordinates with respect to the reference (i.e. bottom center or top center) point, and in each area, the predator selected actions 3, 8, or 12 (every 120 degrees) that will move it toward the reference point, depending on the direction of the prey from the predator's perspective. For instance, if the predator is at (–0.2, 0.8) and the prey is at (–0.2, –0.8), action 12 is selected.

## Behavioral cloning

We constructed neural networks to clone the predatory behavior of rule-based agents. The neural network is composed of two weight layers; that is, it takes the state of environments as inputs as in the deep reinforcement learning agents, processes them through a hidden layer, and then outputs probabilities for each of the 13 potential actions using the softmax function. To ensure a fair comparison with the embedding of deep reinforcement learning agents, we set the number of units in the hidden layer to 32. In the networks, all layers were composed of the fully connected layer. In this study, for each agent (i.e. predator 1 and predator 2), we implemented two types of networks: a linear network without any nonlinear transformation, and a nonlinear network with ReLU activations. Specifically, in the linear network, the hidden layer is composed of the fully connected layer without nonlinearity,

$$h = W_{xh}x + b_h,$$

where $x$, $h$, $W_{xh}$, and $b_h$ denote the input to the hidden layer (state), the output of the hidden layer, the input-to-hidden weight, and the bias, respectively. In the nonlinear network, the hidden layer is composed of the fully connected layers with nonlinearity,

$$h = \varphi(W_{xh}x + b_h),$$

where $\varphi(x)$=max(0, $x$) is the rectified linear unit (ReLU) for nonlinearity. The neural network models were trained to minimize the cross entropy error,

$$E = -\sum_k t_k \log y_k,$$

where $t$ and $y$ denote the actual actions taken by the rule-based agents and the predicted actions in each class $k$, respectively. Network parameters were optimized iteratively using stochastic gradient descent with the Adam optimizer. The learning rate, batch size, and epoch were set as 0.0001, 32, and 2000, respectively, for all agents and networks. The networks were trained, validated, and tested using simulation data for 1000 episodes (123,597 time steps), 100 episodes (16,805 time steps), and 100 episode (12076 time steps), respectively. The network weights were saved according to the best performance observed during the validation phase.

## Data analysis

All data analysis was performed in Python 3.7. Successful predation was defined as the sum of the number of predators catching prey and the number of prey leaving the play area. The theoretical prediction assumes that each predator's performance is independent of the others' performance, and was defined as follows:

$$H_n = 1 - (1 - H_1)^n \tag{11}$$

where $H_n$ and $H_1$ denote the proportion of successful predation when the number of predators is $n$ and 1, respectively. The duration was defined as the time from the beginning to the end of the episode, with a maximum duration of 30 s. The heat maps were constructed based on the frequency of stay in each position, with the play area divided into 1600 (40×40). The concordance rate was calculated by comparing the actual selected action by each agent in the two or three conditions with the action that would be chosen by the agent in the one condition if it were placed in a same situation. The circular correlation coefficient was calculated by converting the selected actions (1–12) into angles (0–330 degrees) (*Berens, 2009*), and in this analysis, action 13 (do nothing) was excluded from the analysis. The two-dimensional embedding was made by transforming the vectors in the last hidden layers of state-value stream and action-value stream in the policy network using t-distributed stochastic neighbor embedding (t-SNE) (*van der Maaten and Hinton, 2008*). To reduce the influence of extremely large or small values, the color ranges of the $V$ value, SD $Q$ value, and distance were limited from the 5th percentile to the 95th percentile of whole values experienced by each agent (see *Figure 3—figure supplements 8–11*).

## Statistics

All quantitative data are reported as mean ± SEM across random seeds in the computational experiment and across participants in the human experiment. In the human experiment, sample sizes were not predetermined statistically, but rather were chosen based on field standards. The data were analyzed using one- or two-way repeated-measures analysis of variance (ANOVA) as appropriate. For these tests, Mauchly's test was used to test sphericity; if the sphericity assumption was violated, degrees of freedom were adjusted by the Greenhouse–Geisser correction. To adjust the $p$ values for multiple comparisons, the Holm-Bonferonni method was used. The data distribution was assumed to be normal for multiple comparisons, but this was not formally tested. Two-tailed statistical tests were used for all applicable analyses. The significance level was set at an alpha value of 0.05. The theoretical prediction was excluded from statistical analyses (*Figures 2a and 4a*) because, from the equation, it is obvious that the proportion of successful predation increases as the number of predators increases. Specific test statistics, $p$ values, and effect sizes for the analyses are detailed in the corresponding figure captions. All statistical analyses were performed using R version 4.0.2 (The R Foundation for Statistical Computing).

## Code availability

The code for computational simulation and figures is available at https://github.com/TsutsuiKazushi/collaborative-hunting; (copy archived at *Kazushi, 2023*).

## Acknowledgements

This work was supported by JSPS KAKENHI (Grant Numbers 20H04075, 21H04892, 21H05300, and 22K17673), JST PRESTO (JPMJPR20CA), and the Program for Promoting the Enhancement of Research Universities.

## Additional information

### Funding

| Funder | Grant reference number | Author |
| --- | --- | --- |
| Japan Society for the Promotion of Science | 20H04075 | Keisuke Fujii |
| Japan Society for the Promotion of Science | 21H04892 | Kazuya Takeda |
| Japan Society for the Promotion of Science | 21H05300 | Keisuke Fujii |
| Japan Society for the Promotion of Science | 22K17673 | Kazushi Tsutsui |
| Japan Science and Technology Agency | JPMJPR20CA | Keisuke Fujii |

The funders had no role in study design, data collection and interpretation, or the decision to submit the work for publication.

### Author contributions

Kazushi Tsutsui, Conceptualization, Resources, Software, Formal analysis, Funding acquisition, Investigation, Visualization, Methodology, Writing - original draft, Writing - review and editing; Ryoya Tanaka, Validation, Investigation, Methodology, Writing - original draft, Writing - review and editing; Kazuya Takeda, Supervision, Funding acquisition, Project administration; Keisuke Fujii, Conceptualization, Supervision, Funding acquisition, Validation, Methodology, Writing - original draft, Project administration, Writing - review and editing

### Author ORCIDs

Kazushi Tsutsui (i) http://orcid.org/0000-0003-3443-0749
Ryoya Tanaka (i) http://orcid.org/0000-0002-6047-6030
Keisuke Fujii (i) http://orcid.org/0000-0001-5487-4297

### Ethics

Human subjects: This study was approved by the Ethics Committee of Nagoya University Graduate School of Informatics. Informed consent was obtained from each participant before the experiment.

### Decision letter and Author response

Decision letter https://doi.org/10.7554/eLife.85694.sa1
Author response https://doi.org/10.7554/eLife.85694.sa2

## Additional files

### Supplementary files

• MDAR checklist

### Data availability

The data and models used in this study are available at https://doi.org/10.6084/m9.figshare.21184069.v3.The code for computational simulation and figures is available at https://github.com/TsutsuiKazushi/collaborative-hunting (copy archived at *Kazushi, 2023*).

The following dataset was generated:

| Author(s) | Year | Dataset title | Dataset URL | Database and Identifier |
|-----------|------|---------------|-------------|-------------------------|
| Tsutsui K, Tanaka R, Takeda K, Fujii K | 2023 | Dataset | https://doi.org/10.6084/m9.figshare.21184069 | figshare, 10.6084/m9.figshare.21184069 |

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
