## [Editor Report]

Cooperative hunting is typically attributed to certain mammals (and select birds) which express highly complex behaviors. This paper makes the valuable finding that in a highly idealized open environment, cooperative hunting can emerge through simple rules. This has implications for a reassessment, and perhaps a widening, of what groups of animals are believed to manifest cooperative hunting.

---

## [Decision Letter]

**Decision letter after peer review:**

Thank you for submitting your article "Collaborative hunting in artificial agents with deep reinforcement learning" for consideration by *eLife*. Your article has been reviewed by 3 peer reviewers, one of whom is a member of our Board of Reviewing Editors, and the evaluation has been overseen by Michael Frank as the Senior Editor.

Essential revisions (for the authors):

1) There are extensive remarks from the reviewers on the validity of some of the more theoretical claims. These need to be carefully addressed. Examples include whether it is correct to infer that more complex cognition is not needed for cooperative hunting because DQN can solve the problem; The absence of an explicit model does not mean that there isn't an implicit model of the other agents behaviors that is encoded in the neural network weights.

2) The lack of causal analysis either needs to be addressed or the "mediated by"-type claims need to be tempered.

3) There has been a translation of the DQN into simple rules, but at present the discussion is too incomplete for readers to understand why this was done and what can be concluded.

4) A discussion of the limitations of this work in terms of the absence of things like shared value functions increasingly common in multi-agent RL and absence of partially observable environments common in predator-prey dynamics, would be helpful.

5) Please address R3's comment that the paper's analysis is possible without RL, such as via behavioral cloning.

6) Some of the reviews suggest better coverage of the relevant literature. Given the size of this literature, this has to be selective, but it appears some effort could be expended to further improve the scholarship of the work.

*Reviewer #1 (Recommendations for the authors):*

Figure 2 – The gap between individual and shared only occurs at two predators, for both % successful predation and duration. It would be helpful to have a discussion of this point. Wild dog packs, for example, are typically larger: perhaps this because the space they work over is much larger (relative to disk size), the environment complexity, or something else, but in any case it would be interesting to know whether shared vs individual is the ruling condition.

It also suggests that sharing is not needed in the 3 predator situation to obtain the same results. Does that mean that the work is suggesting cooperation even occurs without sharing? This seems to be a significant problem, since it's hard to imagine how the term "collaboration" or "cooperation" can be applied in the absence of shared reward. If it is strictly a matter of reduction of duration and increase in rate of success, it may equate to a more limited form of cooperation? Are their biological analogs of group hunts without sharing?

186 – We found that the mappings resulting in collaborative hunting were mediated by distance-dependent internal representations.

'Mediated' here seems to play the role of a "filler term" as used in neuroscience (see Krakauer et al. 2017 Neuroscience needs behavior).

Only correlations have been shown, but this is a causal claim. To support the causal claim, it would be necessary to intervene in the network and show that the interventions in the internal representations have the predicted causal role.

194 – The organization of this paragraph might be better reversed. One could argue that Figure S8 (which could be referenced here) providing similar results and DQN helps support the hypothesis that the representation of distance in the network plays a causal role in the outcome.

234 – Initial position of each episode is unclear as previously noted.

236 – The text above says -1 for moving out of arena to the prey, so if the prey moves out, is it just that, or does the predator also get +1 since the predator is now deemed "successful"?

*Reviewer #2 (Recommendations for the authors):*

– I am not clear that there is sufficient evidence for lines 172-174. The absence of an explicit model does not mean that there isn't an implicit model of the other agents behaviors that is encoded in the neural network weights. I'm not clear what sort of experiment would allow you to distinguish this though there might be a way to run a linear probe to confirm that this information is not in the network weights?

– The notation in lines 246-251 is confusing because it alternates between POMDP notation (i.e. that the agent gets an observation that is a transformation of the true state) and MDP notation. Is the setup an MDP or a POMDP?

– Perhaps line 259 should be "by finding a fixed point of the Bellman equations?"

– Line 271 should be "Dueling Networks" not "Dueling Network" – The sentence starting on line 271 and ending on 273 could or should be cut entirely as it doesn't provide much value and I think it's debatable whether DQN was the first algorithm to solve a high dimensional input problem; it very much depends on how you define high dimensiona

– To get equation 3 from equation 2, there needs to be a factor of 1/2 somewhere.

– In line 321 I don't know what identifiability means in the context of Q-learning? Is this a technical term used in some subfield that works on Q-learning? Why does subtracting the mean help with "identifiability?""

– A discount factor of 0.9 is a wildly low discount factor, basically leading agents to only care about the next 10 steps. I don't think this necessarily affects the outcome of your project or necessarily requires any changes as I don't think agents need to do long horizon reasoning here, but it's worth keeping in mind!

– I don't fully understand the claim that this expands the range of things that are understood to be possible to learn via associative learning. There's no theory precluding a model-free algorithm from learning this type of behavior so the claim in the discussion strikes me as odd. In practice, this type of result where model-free RL agents successfully hunt together have been around since the release of the multi-particle envs (see https://proceedings.neurips.cc/paper/2017/hash/68a9750337a418a86fe06c1991a1d64c-Abstract.html)

– I think the rule-based model is neat but I don't understand what what question it answers. Did I perhaps miss something?

– I don't find the evidence for the distance-dependent features compelling; is all of the evidence for it the t-SNE embeddings?

– Lines 194-196 are confusing to me. Why does there being a rule-based model employ your DQN agent is also learning a similar rule-based model?

*Reviewer #3 (Recommendations for the authors):*My largest suggestion is to fit a linear model to rule-based behaviors and compare the t-SNE embeddings of the behavioral cloning policy with the embeddings of the RL policy? Is the use of RL truly important for this paper?Around line 362, the idea of Rule based agents and human controlled agents are also introduced. I would like to see linear models that take as observations the rule-based agents observations and output the rule based agents actions. Would the t-SNE embeddings look similar for these linear models and for the RL-trained models? If the embeddings look similar, what does that say about the emergence of these capabilities as a result of RL? Does training via RL even matter? Do we care if it doesn't matter?

There is a large amount of work on multi-agent learning that this paper seemingly ignores, or fails to evaluate against. Multi-Agent Actor-Critic for Mixed Cooperative-Competitive Environments has thousands of citations. However, I am willing to accept that there are limitations to what a single paper can cover.

More specific comments:

Line 46-47. I do not know what "simple descriptions with the distances can reproduce similar behavior" is trying to convey.

Lines 50-51: "Our approach of computational ecology bridges the gap between ecology, ethology, and neuroscience and may provide a comprehensive account of them." This is probably too strong of a claim.

Figure 1: The architecture diagram is a little difficult to understand. The key has "layer with ReLU" but then I do not see any clear white boxes? I also do not see any clear units? I think that maybe this is happening inside of the "prey," "predator 1," etc boxes. However, this is all much too small. I think you should decide if you want this figure to be about the neural network architecture, or about the fact the environment is broken into 1 prey and N predators that share an observation.

I think the actions are also not clear. There are probably too many lines in the figure.

For Figure 1 (b), why not just plot the actual density? Actually, I see this is included in Figure 2. I think this is the more helpful Figure!

In Figure 2, what form of Hypothesis testing was used? Was this a KS test? You can't assume the distributions are Gaussian? The presence of a chi-squared statistic seems to indicate Gaussian assumptions. But the distribution is strongly non-Gaussian in this case. A little more clarity would be helpful.

Line 132 mentions that the variance is higher over the action distribution when the predator is about to catch the prey? This is actually the exact opposite of my intuition. I think that the actions hardly matter when the prey is far away, so there is no obvious optimal action, and the choice would be closer to uniform. I'm not sure that this matters very much, but it's interesting.

Line 325 – Usually IL is reserved for Imitation Learning. I have never seen it used for Independent Learning.

Line 324 – I think biological organisms usually model the behavior of other organisms and account for it while planning.

Line 212 – Q-values do implicitly model the competencies of other agents.

Line 196 – What does it mean to switch the decision rules with the distances?

Overall, I think the problems considered by this paper are interesting. And I am happy you took the time to write it. This work made me think a lot about my own research. I appreciate your efforts here. Thank you.

---

## [Author Response]

Essential revisions (for the authors):1) There are extensive remarks from the reviewers on the validity of some of the more theoretical claims. These need to be carefully addressed. Examples include whether it is correct to infer that more complex cognition is not needed for cooperative hunting because DQN can solve the problem; The absence of an explicit model does not mean that there isn't an implicit model of the other agents behaviors that is encoded in the neural network weights.

Thank you for your constructive criticism regarding the validity of our theoretical claims. We have carefully considered the extensive remarks from both you and the reviewers, and have engaged in in-depth discussions with field experts, including a leading scientist on chimpanzee cognition, to address them. As you and the reviewers have pointed out, it is challenging to verify whether an implicit model of other agents’ behaviors is encoded within the neural network weights. Therefore, we have revised our manuscript to refine our claims to specifically address the absence of aspects of “theory of mind”, as it is certain that the agents in our study do not model or infer the “mental states” of others. Specifically, we have revised the description “high-level cognition such as sharing intentions among predators or modeling competencies of other agents” pointed out by the reviewers to “high-level cognition such as aspects of theory of mind” throughout the manuscript. Although this revision may narrow or moderate our argument, we believe it significantly enhances the precision and accuracy of our discussion. Furthermore, by focusing on aspects of theory of mind, we can create a clear distinction from previous studies that incorporated comparable explicit mechanisms, thus aiding the reader's comprehension of our claims and the future directions of our study. We believe these revisions more accurately address the concerns you and the reviewers have raised and ensure our theoretical claims. We are grateful for the guidance that has helped us in improving the manuscript.

2) The lack of causal analysis either needs to be addressed or the "mediated by"-type claims need to be tempered.

Thank you for your insightful comment regarding the lack of causal analysis. We have carefully considered your critique and agree that a more cautious approach is warranted in the absence of a direct causal analysis. In response, we have tempered our claims throughout the manuscript to reflect this. We have modified any statements that may have implied a stronger causal relationship than is currently supported by the data, ensuring that our descriptions accurately represent the correlative nature of our findings, such as being “related to” or “associated with” specific outcomes.

3) There has been a translation of the DQN into simple rules, but at present the discussion is too incomplete for readers to understand why this was done and what can be concluded.

We appreciate your feedback on the translation of the DQN into simple rules and the need for a more comprehensive discussion to aid reader comprehension. As mentioned above, we have revised our manuscript to focus our claims on the absence of aspects of theory of mind. Additionally, in line with the reviewer’s comment, we have revised the sentence in the Introduction section as follows (lines 36 to 39 in the revised manuscript):

“Given that associative learning is likely to be the most widely adopted learning mechanism in animals, collaborative hunting could arise through associative learning, where simple decision rules are developed based on behavioral cues (i.e., contingencies of reinforcement).”

Furthermore, we have added the following sentence to the Introduction section (lines 44 to 46 in the revised manuscript):

“Notably, our predator agents successfully learned to collaborate in capturing their prey solely through a reinforcement learning algorithm, without employing explicit mechanisms comparable to aspects of theory of mind.”

These revisions and additions aim to provide a clearer exposition of why the translation was undertaken and to discuss more explicitly what conclusions can be drawn from it. We hope that these changes will make the ideas more accessible and the underlying reasoning more transparent to our readers.

4) A discussion of the limitations of this work in terms of the absence of things like shared value functions increasingly common in multi-agent RL and absence of partially observable environments common in predator-prey dynamics, would be helpful.

We are grateful for your suggestion to discuss the limitations of our work. In response to your feedback, we have incorporated a discussion of these limitations within the existing conclusion paragraph of our manuscript. In this revision, we have included the “partial observability” alongside other elements of predator-prey dynamics. We have also added the sentence regarding the “shared value functions” as potential directions for future research to suggest areas that could benefit from further exploration. The revised conclusion paragraph is as follows (lines 223 to 234 in the revised manuscript):

“In conclusion, we demonstrated that the decisions underlying collaborative hunting among artificial agents can be achieved through mappings between states and actions. This means that collaborative hunting can emerge in the absence of explicit mechanisms comparable to aspects of theory of mind, supporting the recent idea that collaborative hunting does not necessarily rely on complex cognitive processes in brains. Our computational ecology is an abstraction of a real predator-prey environment. Given that chase and escape often involve various factors, such as energy cost, partial observability, signal communication, and local surroundings, these results are only a first step on the path to understanding real decisions in predator-prey dynamics. Furthermore, exploring how mechanisms comparable to aspects of theory of mind or the shared value functions, which are increasingly common in multi-agent reinforcement learning, play a role in these interactions could be an intriguing direction for future research. We believe that our results provide a useful advance toward understanding natural value-based decisions and forge a critical link between ecology, ethology, psychology, neuroscience, and computer science.”

We believe that these revisions will greatly assist our readers in understanding the scope and implications of our work.

5) Please address R3's comment that the paper's analysis is possible without RL, such as via behavioral cloning.

Thank you for your comment about the possibility of conducting our analysis without the use of RL. We have considered this perspective and have realized that indeed certain analyses could be substituted with behavioral cloning in some cases. Nevertheless, we would like to emphasize that the use of RL brings clarity to several aspects of our study. We describe the reasons for this below from the perspectives of both data and analysis.

Data: Collaborative hunting data is generally scarce, and to our knowledge, no extensive dataset exists with complete locational data on all individuals during hunts. Furthermore, in many cases, obtaining completely controlled data from field observations is challenging. For example, data collected in the wild tend to exhibit some biases (Lang and Farine, 2017). Consequently, we believe that the controlled comparisons presented in our Figure 2 would be difficult to achieve without RL, which makes them notable results.

Analysis: Additionally, we believe that even with a large and controlled dataset, limitations exist in referring the decision-making process from behavioral cloning results and the visualization of internal representations. One such limitation is the prediction accuracy of behavioral cloning, which would not be 100%. Our additional analysis, conducted in response to Reviewer 3's suggestion, demonstrated that prediction accuracy was, at best, 60-70% (Figure 3 supplement 7). In such case, it would be difficult to rule out the possibility that complex cognitive processes are involved in the remaining percentage. Therefore, although we need to be careful in the interpretation of what is being learned on the deep Q-network as you and the reviewers pointed out again and again, the explicit architecture of RL agents strengthens our argument. Thus, even if sufficient data are available, RL would still be meaningful for our analysis.

6) Some of the reviews suggest better coverage of the relevant literature. Given the size of this literature, this has to be selective, but it appears some effort could be expended to further improve the scholarship of the work.

Thank you for pointing that out. Upon reflection, we agree with you and the reviewers that the original manuscript could benefit from more comprehensive coverage of the relevant literature. We have reviewed additional papers focusing mainly on multi-agent reinforcement learning and predator-prey dynamics and have incorporated these into the references in our manuscript.

Reviewer #1 (Recommendations for the authors):Figure 2 – The gap between individual and shared only occurs at two predators, for both % successful predation and duration. It would be helpful to have a discussion of this point. Wild dog packs, for example, are typically larger: perhaps this because the space they work over is much larger (relative to disk size), the environment complexity, or something else, but in any case it would be interesting to know whether shared vs individual is the ruling condition.

We appreciate your comment on the gap between individual and shared condition concerning the proportion of successful predation and duration. We have added a discussion in our manuscript that explores the reasons behind the comparable performance in scenarios involving three predators, whether the prey was shared or not (lines 192 to 199 in the revised manuscript). As you suggested, we considered spatial constraints as a contributing factor to this outcome. Our analysis revealed that predators occasionally exploit the play area's boundaries and the movement of other predators to block the prey's escape path. To illustrate these dynamics, we have added a supplementary figure (Figure 2 supplement 4). We believe this addition will provide a clearer understanding of our result and hope you find this enhancement informative.

It also suggests that sharing is not needed in the 3 predator situation to obtain the same results. Does that mean that the work is suggesting cooperation even occurs without sharing? This seems to be a significant problem, since it's hard to imagine how the term "collaboration" or "cooperation" can be applied in the absence of shared reward. If it is strictly a matter of reduction of duration and increase in rate of success, it may equate to a more limited form of cooperation? Are their biological analogs of group hunts without sharing?

Thank you for your point regarding “cooperation” in the absence of reward sharing. We agree with your comment that the significant improvement in success rates and reduction in hunting duration, compared with the theoretical predictions based on solitary hunting results, suggests a more limited form of cooperation. This concept finds analogous to interspecific group hunting. For example, giant moray eels and groupers have been reported to hunt together though they do not share the prey (Bshary et al., 2006); their repeated interactions may eventually lead to a distribution of prey between both predators. This could be a mutually beneficial relationship that emerges over time without direct reward sharing. Our results also showed a consequent distribution of prey (see Figure 2 supplement 3), suggesting the potential emergence of this form of cooperation.

186 – We found that the mappings resulting in collaborative hunting were mediated by distance-dependent internal representations.'Mediated' here seems to play the role of a "filler term" as used in neuroscience (see Krakauer et al. 2017 Neuroscience needs behavior).Only correlations have been shown, but this is a causal claim. To support the causal claim, it would be necessary to intervene in the network and show that the interventions in the internal representations have the predicted causal role.

Thank you for your constructive comment concerning the use of the term “mediated”. After reviewing the paper by Krakauer et al. 2017 that you referenced, we have understood that our use of “mediated” inappropriately suggested a causal claim in the absence of direct causal analysis. We have therefore revised our manuscript to more accurately reflect the correlational nature of our findings. Specifically, as the editor suggested, we have tempered our statements throughout the manuscript to ensure that it does not imply causality.

194 – The organization of this paragraph might be better reversed. One could argue that Figure S8 (which could be referenced here) providing similar results and DQN helps support the hypothesis that the representation of distance in the network plays a causal role in the outcome.

Thank you for your suggestion regarding the organization of the paragraph. We have revised the text and its sequence to better illustrate the role of the additional analysis with the rule-based model. The revised paragraph reads as follows (lines 200 to 211 in the revised manuscript):

“We found that the mappings resulting in collaborative hunting were related to distance-dependent internal representations. Additionally, we showed that the distance-dependent rule-based predators successfully reproduced behaviors similar to those of the deep reinforcement learning predators, supporting the association between decisions and distances (Methods; Figure 3 supplements 5, 6, and 7). Deep reinforcement learning has held the promise for providing a comprehensive framework for studying the interplay among learning, representation, and decision making, but such efforts for natural behavior have been limited. Our result that the distance-dependent representations relate to collaborative hunting is reminiscent of a recent idea about the decision rules obtained by observation in fish. Notably, the input variables of predator agents do not include variables corresponding to the distance(s) between the other predator(s) and prey, and this means that the predators in the shared conditions acquired the internal representation relating to distance to prey, which would be a geometrically reasonable indicator, by optimization through interaction with their environment. Our results suggest that deep reinforcement learning methods can extract systems of rules that allow for the emergence of complex behaviors.”

234 – Initial position of each episode is unclear as previously noted.

We thank you for your comment on the initial positions of the agents in each episode. We have revised the manuscript to provide a more precise description of the agents’ initial positioning. These details are described in the second paragraph of the Results section and in the Environment subsection of the Methods section (lines 77 to 79, and 247 to 248, in the revised manuscript, respectively).

236 – The text above says -1 for moving out of arena to the prey, so if the prey moves out, is it just that, or does the predator also get +1 since the predator is now deemed "successful"?

We apologize for any confusion caused by the description of the reward and successful predation. During training phase, if the prey moves out of the arena, the predator does not receive a positive reward. We determined that it was not appropriate for the predator to be rewarded in such instances, especially during the early stages of learning, as the movement of the prey outside the arena is often not directly related to the predator's actions. On the other hand, for the evaluation phase, we consider such instances as “successful predation”. This is because, even with trained prey, there are instances where they may exit the arena in an attempt to evade predators, particularly when multiple predators are involved. In such scenarios, it seems reasonable to regard the prey's moving out as indicative of successful predation. To facilitate the reader's understanding, we have added the following clarification to the second paragraph of the Results section (lines 82 to 85 in the revised manuscript):

“During the evaluation phase, if the predator captured the prey within the time limit, the predator was deemed successful; otherwise, the prey was considered successful. Additionally, if one side (predators/prey) moved out of the area, the other side (prey/predators) was deemed successful.”

Furthermore, to avoid any confusion among our readers, we have moved the original description you pointed out from the Environment subsection in the Methods section to the Evaluation subsection (lines 360 to 369 in the revised manuscript). We believe these changes will make the paper more clear and reader-friendly and hope this explanation clarifies your doubts. Thank you for bringing this to our attention.

Reviewer #2 (Recommendations for the authors):– I am not clear that there is sufficient evidence for lines 172-174. The absence of an explicit model does not mean that there isn't an implicit model of the other agents behaviors that is encoded in the neural network weights. I'm not clear what sort of experiment would allow you to distinguish this though there might be a way to run a linear probe to confirm that this information is not in the network weights?

Thank you for your constructive criticism regarding the validity of our theoretical claims. We have carefully considered the feedback from you and the other reviewers, and have engaged in in-depth discussions with field experts, including a leading scientist on chimpanzee cognition, to address them. As you suggested, it is challenging to verify whether an implicit model of other agents’ behaviors is encoded within the neural network weights. Therefore, we have revised our manuscript to refine our claims to specifically address the absence of aspects of “theory of mind”, as it is certain that the agents in our study do not model or infer the “mental states” of others. Although this revision may narrow or moderate our argument, we believe it significantly enhances the precision and accuracy of our discussion. We believe these revisions more accurately address the concerns you and the reviewers have raised and ensure our theoretical claims. We are grateful for the guidance that has helped us to improve the manuscript.

– The notation in lines 246-251 is confusing because it alternates between POMDP notation (i.e. that the agent gets an observation that is a transformation of the true state) and MDP notation. Is the setup an MDP or a POMDP?

We apologize for any confusion caused by the inconsistent notation. The setup we used is an MDP, not a POMDP. We have corrected the relevant descriptions in the Methods section and the notation in Figure 1a to consistently reflect an MDP framework (lines 257 to 263 in the revised manuscript). Thank you for bringing this to our attention, and we appreciate your patience as we rectify this error.

– Perhaps line 259 should be "by finding a fixed point of the Bellman equations?"

Thank you for your suggestion concerning the phrasing. We have amended the manuscript accordingly (line 271 in the revised manuscript). We appreciate your attention to detail and your assistance in enhancing the technical accuracy of our paper.

– Line 271 should be "Dueling Networks" not "Dueling Network"

Thank you for pointing out the correct terminology. We have made the correction to “Dueling Networks” as you suggested (line 283 in the revised manuscript). Additionally, we have capitalized the initial letters of RL methods throughout the manuscript.

– The sentence starting on line 271 and ending on 273 could or should be cut entirely as it doesn't provide much value and I think it's debatable whether DQN was the first algorithm to solve a high dimensional input problem; it very much depends on how you define high dimensiona

We appreciate your critical feedback on the sentence. Upon reflection, we agree with your suggestion and have therefore removed it from the manuscript. Thank you for guiding us towards a more concise and accurate presentation of our work.

– To get equation 3 from equation 2, there needs to be a factor of 1/2 somewhere.

Thank you for pointing out the discrepancy between equations 2 and 3. We have included the factor of 1/2 to ensure the correctness of the equations. Again, we appreciate your attention to detail and your assistance with our work.

– In line 321 I don't know what identifiability means in the context of Q-learning? Is this a technical term used in some subfield that works on Q-learning? Why does subtracting the mean help with "identifiability?""

Thank you for your careful review and for bringing to our attention the term “identifiability”. We referred to the term as it was introduced in the paper by Wang et al. (2016) on Dueling Networks. However, after re-evaluating its usage based on your suggestion, we agree that subtracting the mean does not necessarily aid identifiability. Consequently, we have removed the related sentences from the Methods section of our manuscript and appreciate your guidance on this matter.

– A discount factor of 0.9 is a wildly low discount factor, basically leading agents to only care about the next 10 steps. I don't think this necessarily affects the outcome of your project or necessarily requires any changes as I don't think agents need to do long horizon reasoning here, but it's worth keeping in mind!

Thank you for your advice on the choice of the discount factor. We will certainly take this into consideration and pay close attention to the impact of different discount factors on agent behavior in future research!

– I don't fully understand the claim that this expands the range of things that are understood to be possible to learn via associative learning. There's no theory precluding a model-free algorithm from learning this type of behavior so the claim in the discussion strikes me as odd. In practice, this type of result where model-free RL agents successfully hunt together have been around since the release of the multi-particle envs (see https://proceedings.neurips.cc/paper/2017/hash/68a9750337a418a86fe06c1991a1d64c-Abstract.html)

Thank you for your input on our discussion about associative learning. After considering your perspective, we agree with your comment and have removed the related statements from the discussion.

– I think the rule-based model is neat but I don't understand what what question it answers. Did I perhaps miss something?

The rule-based model was developed to support our claim that predator agents' decisions are related to distance-dependent internal representations. We examined the state vectors of last hidden layers in each agent's network, which lead to action values after a single linear transformation and aggregation. With this in mind, we posited that if neural networks encode the distances between predators and prey, a concise rule-based model based on these distances should replicate similar behaviors. This additional analysis, prompted by a reviewer's comments, sought to substantiate our claim. While successfully replicating predator behavior using a distance-dependent rule-based model does not completely prove that the RL agent's decision are associated with the distances, it would provide support for our assertion. Additionally, in response to Reviewer 1's suggestion, we have relocated the description of these results to a dedicated paragraph in the Discussion section that explores the relationship between agent decisions and distance-dependent representations (lines 201 to 203 in the revised manuscript), thereby clarifying the aim of this additional analysis for the reader.

– I don't find the evidence for the distance-dependent features compelling; is all of the evidence for it the t-SNE embeddings?

As mentioned in our previous response, our assertion that predator agents’ decisions are related to the distances is supported by analyses of rule-based modeling as well as t-SNE embeddings. These approaches aim to provide a comprehensive understanding of the role of the distances in the agents' decision processes. Additionally, as Reviewer 1 highlighted, the distances between predators and prey agents are directly related to their rewards, making it plausible that these distances factor into the computation of action values during decision-making. We believe that this evidence addresses your concerns.

– Lines 194-196 are confusing to me. Why does there being a rule-based model employ your DQN agent is also learning a similar rule-based model?

Thank you for your continued engagement with our work. While partially reiterating what was mentioned in our previous response, we would like to clarify the rationale behind employing a rule-based model. It is to demonstrate that if predator agents encode the distances between predators and prey within their neural networks, these distances could potentially be used to construct a simple rule-based model that replicates the agents’ behavior. We have recognized that the initial presentation of the rule-based model's description could have been abrupt and confusing. Consequently, we have moved this discussion to the fourth paragraph of the Discussion section (lines 200 to 211 in the revised manuscript) and have provided a detailed explanation of its purpose and implementation in the Methods section (lines 396 to 432 in the revised manuscript). This rearrangement aims to make the intent and methodology of the additional analysis clearer to the reader.

Reviewer #3 (Recommendations for the authors):My largest suggestion is to fit a linear model to rule-based behaviors and compare the t-SNE embeddings of the behavioral cloning policy with the embeddings of the RL policy? Is the use of RL truly important for this paper?

Thank you for your substantial suggestion to compare the t-SNE embeddings of both the behavioral cloning policy and the RL policy. As advised, we implemented two types of networks: a linear network without any nonlinear transformation and a nonlinear network with ReLU activations, and have conducted the comparison as shown in Figure 3 —figure supplement 7 (top: linear network, bottom: nonlinear network).

We found the results to be intriguing because these are somehow similar to the RL embeddings, which we consider promising for potential application to other biological data we possess. We are grateful for this insightful recommendation, and details of these procedures and results have been added to Methods section and Supplementary Figures (lines 433 to 452, and Figure 3 supplement 7, in the revised manuscript, respectively). However, as mentioned in our response to the editor, we firmly believe that the use of RL was essential for the outcomes presented in this paper. While some analyses could indeed be substituted with behavioral cloning in some cases, as this additional analysis has shown, we believe that the use of RL is still important for this paper. The reasons for this are described below from the respective perspectives of data and analysis.

Data: Collaborative hunting data is generally scarce, and to our knowledge, no extensive dataset exists with complete locational data on all individuals during hunts. Furthermore, in many cases, obtaining completely controlled data from field observations is challenging. For example, data collected in the wild tend to exhibit some biases (Lang and Farine, 2017). Consequently, we believe that the controlled comparisons presented in our Figure 2 would be difficult to achieve without RL, which makes them notable results.

Analysis: Additionally, we believe that even with a large and controlled dataset, limitations exist in referring the decision-making process from behavioral cloning results and the visualization of internal representations. One such limitation is the prediction accuracy of behavioral cloning, which would not be 100%. Our additional analysis, conducted in response to your suggestion, demonstrated that prediction accuracy was, at best, 60-70% (Figure 3 supplement 7). In such case, it would be difficult to rule out the possibility that complex cognitive processes are involved in the remaining percentage. Therefore, although we need to be careful in the interpretation of what is being learned on the deep Q-network as you and the other reviewers pointed out again and again, the explicit architecture of RL agents strengthens our argument. Thus, even if sufficient data are available, RL would still be meaningful for our analysis.

Around line 362, the idea of Rule based agents and human controlled agents are also introduced. I would like to see linear models that take as observations the rule-based agents observations and output the rule based agents actions. Would the t-SNE embeddings look similar for these linear models and for the RL-trained models? If the embeddings look similar, what does that say about the emergence of these capabilities as a result of RL? Does training via RL even matter? Do we care if it doesn't matter?

As mentioned in our response to your previous comment, we tested both a linear network you suggested and a nonlinear network which is more similar to the RL network. For both networks, we aligned the inputs with those given to the RL network. As demonstrated above, we found that the embeddings do separate to some extent based on the distances. Interestingly, despite the differences in prediction accuracy between the networks, the embeddings were quite similar. These findings suggest the usefulness of analyzing decision-making processes through behavioral cloning, as you have suggested, and show potential for application to the biological data, as already noted. However, as previously mentioned, models with an explicit structure like RL agents bring clarity to our study and are essential in substantiating our arguments. The explicitness of the RL model architecture helps us to dissect and articulate the mechanisms more precisely.

There is a large amount of work on multi-agent learning that this paper seemingly ignores, or fails to evaluate against. Multi-Agent Actor-Critic for Mixed Cooperative-Competitive Environments has thousands of citations. However, I am willing to accept that there are limitations to what a single paper can cover.

We apologize for the oversight and appreciate your pointing out the omission of significant multi-agent learning literature. We understand the importance of situating our work within the broader research context and have now included relevant citations in our manuscript. We regret any impression of neglecting existing contributions. If there are any specific references you believe we should include, we would appreciate being informed.

More specific comments:Line 46-47. I do not know what "simple descriptions with the distances can reproduce similar behavior" is trying to convey.

Thank you for your comment. We have removed the phrase you pointed out “simple descriptions with the distances can reproduce similar behavior” because we deemed it unnecessary. As mentioned in our response to Reviewer 2, the additional analysis was intended to support the insights gained from the t-SNE embedding analyses. To clarify the purpose of this additional analysis, we have relocated the relevant sentences to the paragraph in the Discussion section that deals with the relationship between agent decisions and internal representations (lines 200 to 211 in the revised manuscript). In addition, to improve clarity for the reader, further details have been consolidated in the Methods section and Supplementary Figures (lines 396 to 432, and Figure 3 supplements 5 to 7, in the revised manuscript, respectively).

Lines 50-51: "Our approach of computational ecology bridges the gap between ecology, ethology, and neuroscience and may provide a comprehensive account of them." This is probably too strong of a claim.

Thank you for your critique regarding the claim made in our manuscript. In accordance with your feedback, we have tempered the statement. Instead of suggesting a bridging of gaps across disciplines, we just assert that (lines 52 to 53 in the revised manuscript):

“Our results support the recent suggestions that the underlying processes facilitating collaborative hunting can be relatively simple.”

This revised statement focuses on the contribution of our work to the existing literature by providing evidence that supports current hypotheses about the simplicity of mechanisms underlying collaborative behavior.

Figure 1: The architecture diagram is a little difficult to understand. The key has "layer with ReLU" but then I do not see any clear white boxes? I also do not see any clear units? I think that maybe this is happening inside of the "prey," "predator 1," etc boxes. However, this is all much too small. I think you should decide if you want this figure to be about the neural network architecture, or about the fact the environment is broken into 1 prey and N predators that share an observation.

Thank you for your feedback on Figure 1. In line with your suggestions, we have revised the figure to better highlight the environmental setup involving one prey and N predators, thereby aiming to enhance readability and comprehension for readers. Specifically, we have removed the legends “unit”, “layer with ReLU”, “forward connection”, and “aggregating module”. Furthermore, for readers interested in a visualization of the network, we have referenced the Supplementary Figure (Figure 1 supplement 1) that illustrates the network architecture in the “Training details” subsection of the Methods section (line 342 in the revised manuscript).

I think the actions are also not clear. There are probably too many lines in the figure.

Thank you for your comment regarding the indication of actions in Figure 1a. The agents in our study can perform a total of 13 actions: acceleration in 12 directions plus an option to do nothing. The illustration in the “Action” part of Figure 1a accurately depicts these with 12 arrows. Your observation about the excess of lines might also relate to the four lines each for “State”, “Reward”, and “Action”, reflecting the independent learning framework employed in our study. Consolidating these lines into a single one could potentially obscure the individual learning processes of the agents. Therefore, while acknowledging that the figure may appear somewhat cluttered, we have opted to keep the distinct lines as they are to maintain clarity and avoid misunderstanding. We hope this clarification addresses your concerns.

For Figure 1 (b), why not just plot the actual density? Actually, I see this is included in Figure 2. I think this is the more helpful Figure!

Thank you for your suggestion regarding Figure 1b. Indeed, we initially created heat maps to represent the data. However, we found that for conditions where episodes ended quickly, that is the fast and equal conditions, the heat maps were predominantly influenced by the initial positions, resulting in a concentration of distribution in the center of the play area. Therefore, we decided to first present trajectories for each condition to capture the general behavior of the agents and then focused on providing a heat map for the slow condition, where the episode duration was longer and less influenced by the initial positions. Following your valuable feedback, we have added heat maps for the fast and equal conditions as Supplementary Figures to accommodate readers interested in visualizing the density across all conditions (Figure 2 supplement 1 in the revised manuscript). We hope this addition will be helpful.

In Figure 2, what form of Hypothesis testing was used? Was this a KS test? You can't assume the distributions are Gaussian? The presence of a chi-squared statistic seems to indicate Gaussian assumptions. But the distribution is strongly non-Gaussian in this case. A little more clarity would be helpful.

We appreciate your detailed attention to the statistical analysis in our manuscript. Based on the context of variability you have described, we believe that your comments refer to Figure 4 rather than Figure 2. Regarding the sample size of 10 per condition, we acknowledge that the central limit theorem may not provide a strong justification for the assumption of normality. However, we would like to emphasize the robustness of ANOVA when dealing with small sample sizes and its ability to yield reliable results even when data distributions deviate from normality. The lack of formal testing for normality is indeed a limitation, as noted in Statistics subsection in the Methods section of our manuscript (lines 469 to 481 in the revised manuscript). Yet, the ANOVA test has been widely recognized for its robustness, especially in the context of balanced designs, which is the case in our experimental setup. Moreover, the Holm-Bonferroni method has been applied to adjust for multiple comparisons, reducing the risk of Type I errors. We believe that these considerations, along with the conservative nature of our statistical correction methods, provide a reasonable basis to uphold the validity of our findings. Our approach aligns with common practices in the field, where the practical constraints of sample collection and experimental design often necessitate a balance between statistical ideals and real-world applicability.

Line 132 mentions that the variance is higher over the action distribution when the predator is about to catch the prey? This is actually the exact opposite of my intuition. I think that the actions hardly matter when the prey is far away, so there is no obvious optimal action, and the choice would be closer to uniform. I'm not sure that this matters very much, but it's interesting.

Thank you for your comment regarding the variance in action values. We appreciate this opportunity to clarify the interpretation of the variance of action values in our study. A larger variance in action values indicates a situation where there is a significant distinction in the value of possible actions, with some actions being highly valued and others much less so. Conversely, a smaller variance suggests that there is little difference in the value of actions, with the distribution of action values being closer to uniform. This can be observed in Figure 3 supplement 4 in the revised manuscript, where the action values of predator 2 indeed approach a uniform distribution when the prey is distant. Therefore, it seems our findings are consistent with your intuition that when the prey is far away, the actions matter less. If there is any misunderstanding, we would be grateful for the opportunity to ensure our interpretations align with the observed behavior.

Line 325 – Usually IL is reserved for Imitation Learning. I have never seen it used for Independent Learning.

Thank you for bringing this to our attention. We have removed the abbreviation “IL” for Independent Learning to avoid any confusion.

Line 324 – I think biological organisms usually model the behavior of other organisms and account for it while planning.

Thank you for your insightful comment regarding the modeling of biological behaviors. Our initial intention was to illustrate that each policy network in our computational model operates independently, similar to individual neural processes in biological brains, without the shared network weights that are often used in multi-agent reinforcement learning environments. Nonetheless, we agree with your observation that the lack of explicit mechanisms for modeling and planning may not entirely reflect the intricacies of biological organisms. Consequently, we have revised the manuscript to remove the term “biologically plausible” from the Results and Methods sections to prevent any overstatement of our computational model's capabilities. The revised description is as follows (lines 331 to 332 in the revised manuscript):

“We here modeled an agent (predator/prey) with independent learning, one of the simplest approaches to multi-agent reinforcement learning.”

We believe this modification more accurately conveys our methodology and the scope of our study.

Line 212 – Q-values do implicitly model the competencies of other agents.

Thank you for your point out. As you and the other reviewers noted, we have recognized that deep Q-networks implicitly model the competencies of other agents. Therefore, we have revised our manuscript to refine our claims to specifically address the absence of aspects of “theory of mind”, as it is certain that the agents in our study do not model or infer the “mental states” of others. Although this revision may narrow or moderate our argument, we believe it significantly enhances the precision and accuracy of our discussion. We are grateful for the guidance that has helped us to improve the manuscript.

Line 196 – What does it mean to switch the decision rules with the distances?

Thank you for your comment. We have removed the phrase you pointed out “switch the decision rules with the distances” because we deemed it ambiguous as you suggested. As mentioned in our response to your previous comment, we have rearranged the relevant paragraph in the Discussion section to clarify the purpose of this additional analysis (lines 200 to 211 in the revised manuscript). We are grateful for the guidance that has helped us to improve the manuscript.

Overall, I think the problems considered by this paper are interesting. And I am happy you took the time to write it. This work made me think a lot about my own research. I appreciate your efforts here. Thank you.

Thank you for your positive feedback. We are delighted to hear that our paper has sparked further thought about your research. It is encouraging to know that the problems we have addressed are considered interesting within the research community. Your kind words are greatly appreciated.